# A Lévy expansion strategy optimizes early dune building by beach grasses

Valérie C. Reijers [1], Koen Siteur[2,3], Selwyn Hoeks[1,4], Jim van Belzen [3,5], Annieke C.W. Borst[1], Jannes H.T. Heusinkveld[6], Laura L. Govers[1,7], Tjeerd J. Bouma[3,7,8], Leon P.M. Lamers[1], Johan van de Koppel[3,7] & Tjisse van der Heide[1,7,9]

Lifeforms ranging from bacteria to humans employ specialized random movement patterns. Although effective as optimization strategies in many scientific fields, random walk application in biology has remained focused on search optimization by mobile organisms. Here, we report on the discovery that heavy-tailed random walks underlie the ability of clonally expanding plants to self-organize and dictate the formation of biogeomorphic landscapes. Using cross-Atlantic surveys, we show that congeneric beach grasses adopt distinct heavy-tailed clonal expansion strategies. Next, we demonstrate with a spatially explicit model and a field experiment that the Lévy-type strategy of the species building the highest dunes worldwide generates a clonal network with a patchy shoot organization that optimizes sand trapping efficiency. Our findings demonstrate Lévy-like movement in plants, and emphasize the role of species-specific expansion strategies in landscape formation. This mechanistic understanding paves the way for tailor-made planting designs to successfully construct and restore biogeomorphic landscapes and their services.

[1] Department of Aquatic Ecology & Environmental Biology, Institute for Water and Wetland Research, Radboud University, Faculty of Science, Heyendaalseweg 135, Nijmegen, AJ 6525, The Netherlands. [2] Shanghai Key Laboratory for Urban Ecological Processes and Eco-Restoration & Center for Global Change and Ecological Forecasting, School of Ecological and Environmental Science, East China Normal University, 200241 Shanghai, China. [3] Department of Estuarine and Delta Systems, Royal Netherlands Institute for Sea Research and Utrecht University, Yerseke, NT 4401, The Netherlands. [4] Department of Environmental Science, Institute for Water and Wetland Research, Radboud University, Faculty of Science, Heyendaalseweg 135, Nijmegen, AJ 6525, The Netherlands. [5] Ecosystem Management Research Group, University of Antwerp, Wilrijk 2610, Belgium. [6] The Fieldwork Company, Groningen, GV 9721, The Netherlands. [7] Conservation Ecology Group, Groningen Institute for Evolutionary Life Sciences, University of Groningen, Groningen, CC 9700, The Netherlands. [8] Faculty of Geosciences, Department of Physical Geography, Utrecht University, Utrecht, TC 3508, Netherlands. [9] Department Coastal Systems, Royal Netherlands Institute for Sea Research and Utrecht University, Den Burg, AB 1790, The Netherlands. Correspondence and requests for materials should be addressed to V.C.R. (email: v.reijers@science.ru.nl)

In the quest for food, shelter, or conspecifics, mobile organisms such as bacteria, mussels, birds, fish, and even humans have been found to employ specialized search strategies that are well-described by various types of random walks[1–6]. The simplest and most commonly observed form, the Brownian walk, yields a single densely-spaced search path by following an exponential step size distribution with mostly small steps. However, an increasing number of studies reports clear deviations from this simple strategy, in which organisms adopt alternative movement patterns characterized by heavy-tailed step size distributions that include incidental large steps. The archetypical example of such a strategy is the scale-invariant Lévy walk, which generates a power-law distribution of small localized search paths interspaced with larger steps. Lévy walks have been suggested to optimize search success when resources are sparse and erratically distributed[1,7–9]. Although successfully used as optimization strategies in many scientific fields[6,10,11], application of random walks in biology has remained focused on the realm of search optimization by mobile organisms.

In this study, we demonstrate that heavy-tailed random walk strategies underlie the ability of plants to control the formation of biogeomorphic landscapes. Such organism-engineered systems, which include river delta's, salt marshes, coastal dunes and seagrass meadows, generate over 10 trillion US$ annually in ecosystem services, such as flood protection, water purification, nutrient cycling, carbon storage, tourism enhancement and sustainment of biodiversity[12–16]. Recent work revealed that the formation of biogeomorphic landscapes critically depends on the ability of landscape-building clonal plants to successfully establish by creating sufficiently large vegetation patches that are essential

to initiate self-promoting feedbacks[17–19]. Clonally expanding plants stimulate sedimentation of airborne and water-suspended particles with increasing patch size and shoot density, which promotes their own growth and survival[18–20]. An important drawback of tight shoot clustering, however, is that landscape colonization becomes relatively slow[21]. Whereas the importance of both rapid colonization and the initiation of landscape-building feedbacks is now well-recognized[13], it remains unknown if colonizing landscape-forming plants spatially organize their shoots to combine the needs for tight patch formation and clonal expansion. Here, we hypothesize that colonizing coastal plants employ a Lévy-type expansion strategy to create a clonal network consisting of multiple dense shoot patches that maximize self-promoting feedbacks at the landscape scale with a minimum investment in covering distances.

To test our hypothesis, we investigated how colonizing dune-building grasses organize their shoots to initiate dune building. Vegetated coastal dunes protect about one-third of the world's shorelines[20,22]. However, the size and shape of these dunes and thus their ability to defend the hinterland can differ greatly depending on the dune-building species involved[23]. For instance, *Ammophila arenaria* (European marram grass) forms tall and steep dunes, whereas dunes formed by its North American congener, *Ammophila breviligulata* (American beachgrass) are much lower and wider and therefore considered less effective in protecting the hinterland—even when growing in the same environment (Fig. 1)[23–25]. In addition, the plants differ in their physiological tolerance to burial and flooding stress, respectively, with *A. arenaria* being more resistant to burial stress by developing vertically expanding rhizomes, while *A. breviligulata* has a

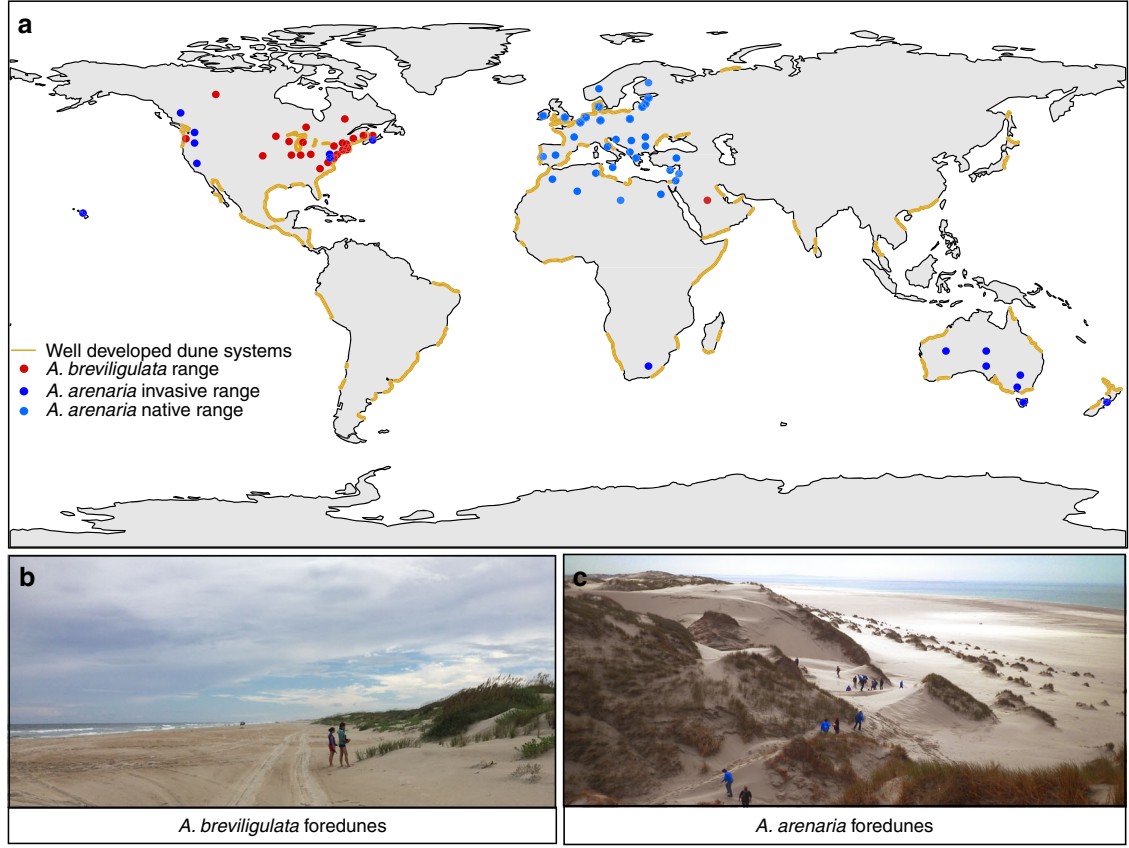

**Fig. 1** Distribution and dune morphology of both *Ammophila* species. **a** map showing worldwide distribution of well-developed dune systems and the occurrence of both *Ammophila* species (adapted from refs. [22,58]). **b** A typical low and wide foredune dominated by *A. breviligulata* (photo: V. Reijers), and **c** a typical tall and steep foredune dominated by *A. arenaria* (photo: N. van Rooijen)

higher salinity tolerance. This suggests that both species have adopted different dune-building strategies to cope with the stressful conditions of growing at the land-sea interface[26]. So far, studies on the biophysical feedback strength of the two species have related observed differences in dune morphology to species-specific differences in shoot densities in existing dune fields and their growth response to sand burial. Specifically, they conclude that (i) higher shoot densities promote sand capture with *A. arenaria* typically generating more shoots per square metre than *A. breviligulata* in existing dune fields, and (ii) the shooting rate of *A. arenaria* is stronger stimulated by sand capture compared with *A. breviligulata*. Yet, it remains to be elucidated whether dune-building grasses control biophysical engineering strength via the spatial arrangement of their shoots in the beach colonization phase when initiating dune formation is vital for escaping physical stress from flooding. Using random search models we aim to unravel (i) whether dune-building species differ in their clonal expansion strategy and (ii) whether the observed expansion strategies and the resulting spatial shoot organizations can be related to the sand-trapping potential in these early phases. Our study shows that dune-building grasses have adopted different clonal expansion strategies to optimize their engineering strength during the early phase of beach colonization. These findings expand the application of heavy-tailed random walk models in biology and call for adaptive restoration schemes that take the spatial organization of landscape-forming plants into account.

## Results

**Species-specific clonal strategies affect shoot organization.** We first investigated what type of clonal expansion strategy was employed by *A. arenaria* along the Dutch North Sea coast and by *A. breviligulata* along the eastern US coast, respectively. To study their clonal expansion process in the early phase of establishment (0.5–1.5-year-old plants), we selected isolated plants growing at the foot of the dunes. First indirect support for our hypothesis was provided by analysing the spatial shoot organization of expanding *A. arenaria* and *A. breviligulata* plants. Spatial cluster analyses revealed that both species strongly deviated from a

homogeneous distribution, with *A. arenaria* exhibiting a shoot organization with a fractal dimension of 0.8 over a range of values that our sampling method allowed (4–16 cm) (Supplementary Figure 2). Since point patterns generated by Lévy movement generally lack a specific scale (Lévy dust)[27,28], this provided a first indication that beach grasses seem to diverge from simple Brownian movement processes and follow more complex Lévy-like expansion strategies[29].

We further investigated whether the spatial shoot patterns can be used as a signature for their clonal expansion strategy by reconstructing the rhizomal network of both species. To estimate step sizes between individual shoots within the clonal network, we applied a simple connecting algorithm (Nearest Neighbour search), validated by excavation of the rhizomal networks, to images with mapped coordinates of all shoots (see Methods section). Results revealed that the expansion strategies—as defined by the step size distribution—of both species clearly deviate from a simple Brownian strategy and are better described by heavy-tailed step size models such as a Lévy or a Composite Brownian walk (Fig. 2) (see Methods section for detailed description of fitting procedure). Specifically, the step size distribution of *A. arenaria* was best described by a truncated Lévy distribution with a power-law exponent ($\mu$) of 1.98, while *A. breviligulata* was best approximated by a Composite Brownian distribution that closely matched a truncated Lévy distribution with $\mu = 1.5$ (Fig. 2, see Supplementary Figure 5 for a visual representation of all fitted distributions per species). The findings on the combined step data were consistent with analyses of individual plants, where Lévy or truncated Lévy distributions best described 83% of the *A. arenaria* individuals, while Composite Brownian was the best-supported model for most of the *A. breviligulata* plants (75%) (Supplementary Figure 5 and Supplementary Table 1). Notably, the Lévy or power-law exponent obtained for *A. arenaria* ($\mu = 1.98$) is close to the theoretical optimum of a Lévy walk at $\mu = 2$[7], which emerges as a trade-off between the tendency of moving away and intensive searching and generates a fractal patchy shoot pattern (i.e. Lévy dust), whereas *A. breviligulata* ($\mu = 1.5$) forms a more dispersed shoot organization (i.e. a larger proportion of longer steps).

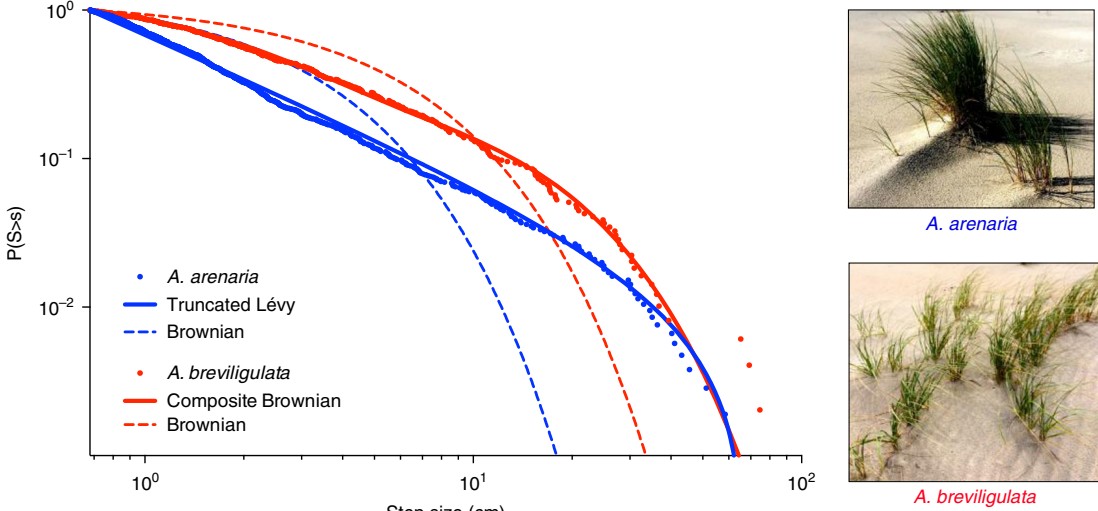

**Fig. 2** The clonal expansion strategy of both *Ammophila* species. Inverse cumulative frequency distribution of the pooled step size data (step size >0.68 cm) obtained for both *Ammophila arenaria* (1053 steps from eight individual plants) and *Ammophila breviligulata* (492 steps from four individual plants). The dashed lines represent the best-fitted exponential distribution (Brownian) for *A. arenaria* (blue) and *A. breviligulata* (red), respectively. The best fit for the total data set, based on weighted AIC value (see Supplementary Table 1), was a truncated Lévy (blue line) for *A. arenaria* and a two-mode Composite Brownian (red line) for *A. breviligulata*. Source data are provided as a Source Data file

**Clonal expansion strategy determines sand-trapping**. To further test our hypothesis that a Lévy-like step size distribution optimizes sand-trapping potential in the colonization phase, we developed a spatially explicit 2D model that simulates the cumulative effect of individual shoots on wind speed (see Methods section). In this minimal model, we described clonal shoot expansion as a random walk process, and manipulated spatial shoot organization by varying the power-law exponent $\mu$ of a truncated Lévy distribution from which the step sizes were drawn. Specifically, we gradually shifted $\mu$ from 1.1 (~Ballistic), via 2.0 (Lévy optimum), to 3.0 (~Brownian). For each step, we then simulated wind flow over the grid and determined the potential area of sand deposition by presuming that deposition occurs when wind speed is reduced below a critical threshold (see Methods section). Simulations revealed that the clonal expansion strategy is a strong determinant of the sand-trapping capacity of dune grasses, with a more dispersed Lévy-type strategy ($\mu \sim 1.5$)

yielding the highest cumulative area of sand deposition (Fig. 3d). The outcome changes when accounting for the relatively high energy investment of this dispersed strategy, which requires covering long distances relative to more clumping strategies ($\mu > 2$) (Fig. 3d). Collected field data suggest that resource efficiency is critical for plants growing in these sandy systems, as the data revealed very low nutrient levels in the soils and leaf tissue of both species (Supplementary Table 2). When we express sand-trapping efficiency as the area with potential sand deposition per unit effort, i.e. the average rhizome length the plant grows between shoots, we find that the patchy Lévy strategy associated with $\mu \sim 2$ becomes most efficient.

Additional analyses demonstrate that this effect becomes increasingly apparent as the number of shoots in the clonal network increases, although the number of shoots required depends on wind conditions (Supplementary Figure 7, Supplementary Table 3). These results demonstrate the saturating effects

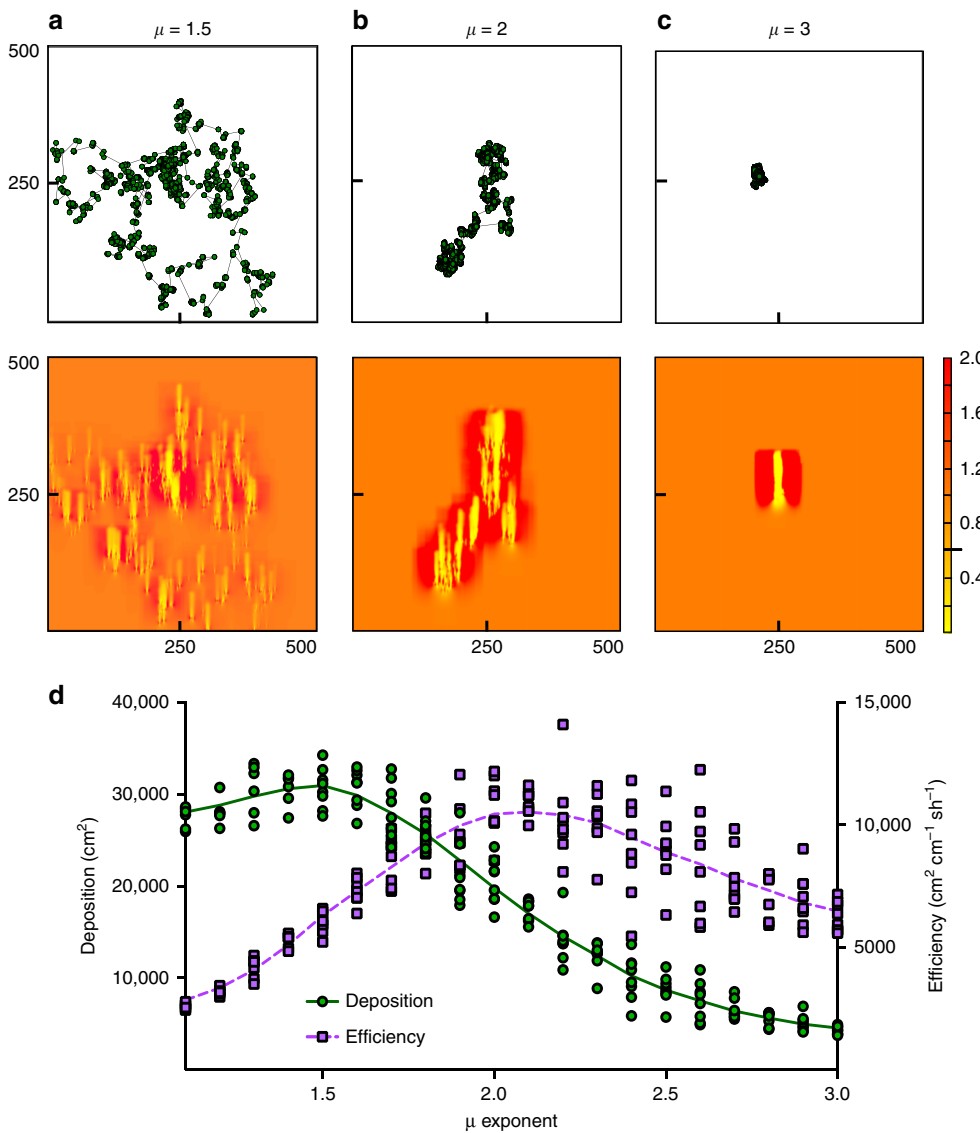

**Fig. 3** Effect of clonal expansion strategy on the sand-trapping capacity. **a–c** Model results showing the effect of the step size distribution (dispersed, $\mu \sim 1.5$; Lévy, $\mu \sim 2$; Brownian, $\mu \sim 3$) on wind speed profiles for a clonal network consisting of 4000 shoots ($N = 8$, scale on the panel figures is in cm). The black indicator on the scale bar at 0.61 indicates the threshold fraction of the wind speed below which sand is deposited. **d** Sand deposition is highest for the more dispersed strategy and decreases with increased clustering of shoots (green line, left axis). Sand-trapping efficiency, calculated as sand deposition divided by the average rhizome length between shoots, was highest at the Lévy optimum of $\mu \sim 2$ (dashed purple line, right axis). Error bars represent ± s.e. m. Source data are provided as a Source Data file

a clumping strategy ($\mu > 2$) may have on potential sand capture. It therefore highlights why an intermittently clumped Lévy-like strategy ($\mu \sim 2$) in early colonization phases (<100 shoots) leads to high potential sand deposition, but on the long run is outcompeted by a more dispersed strategy ($\mu \sim 1.5$) (Supplementary Table 3). Similarly, a highly clumped strategy ($\mu \sim 3$) is more efficient when shoot numbers are low, but as the plant grows, the added attenuating effect of shoots on wind flow decreases due to overlap. Hence, the heavy-tailed Lévy-like strategy of $\mu \sim 2$, as observed for *A. arenaria*, becomes more efficient over time by generating multiple shoot patches that maximize engineering effects, while simultaneously colonizing a large area with minimum investment in covering distances.

**Experimental validation of biophysical feedback.** Finally, to test our model findings under natural wind conditions in the field, we conducted an experiment in which we manipulated the spatial organization of dune grasses using artificial shoot mimics. Specifically, we constructed plots of 4 m² in which we planted the same number of mimics (~2000 shoots) in three spatially distinctive patterns: dispersed (representing a more ballistic strategy), patchy (representing a Lévy-like strategy) and single-patch (representing a more Brownian strategy) (Fig. 4c–e). Our experimental results were consistent with the model findings. Sand capture, represented by the total volume of sand, was the highest in the dispersed pattern (Fig. 4a). For sand-trapping efficiency, however, we found the patchy (Lévy-like) pattern to outperform the other treatments by at least two times (Fig. 4b).

## Discussion

Our work provides compelling evidence that landscape-building plants such as beach grasses apply heavy-tailed clonal expansion strategies in the early colonization phase. In contrast to a simple Brownian strategy that yields a single dense patch of shoots, these strategies generate more patchy shoot organizations that balance the need for local modification and area colonization. Specifically, we found that the Lévy-like strategy of *A. arenaria* maximizes sand-trapping efficiency by accreting sediment within multiple dense shoot patches, while the more dispersed strategy of *A. breviligulata* maximizes total sand capture over a wider area. Previous studies found that, although *A. breviligulata* is generally regarded the stronger competitor, *A. arenaria* can prevail under low sand supply[23,30]. The Lévy-type expansion of *A. arenaria* may explain its efficiency in sand-limited environments, as this strategy may prevent sediment depletion by accreting sand within shoot patches rather than distributing it over a wider area. In contrast, the more dispersed *A. breviligulata* strategy accretes sand over a wider area, preventing local detrimental effects of excessive sand burial. Overall, our work builds on previous studies suggesting that differential growth strategies can help explain the emergence of contrasting dune morphologies[23,31,32], by demonstrating that beach grasses adopt distinct colonization strategies that determine their engineering strength in these early developmental stages. Once these plants have successfully established, coastal dune formation is then further steered by biophysical feedbacks between sediment supply, growth response of vegetation to sediment accumulation and the rate of disturbances that negatively impact vegetation survival[20,24,32].

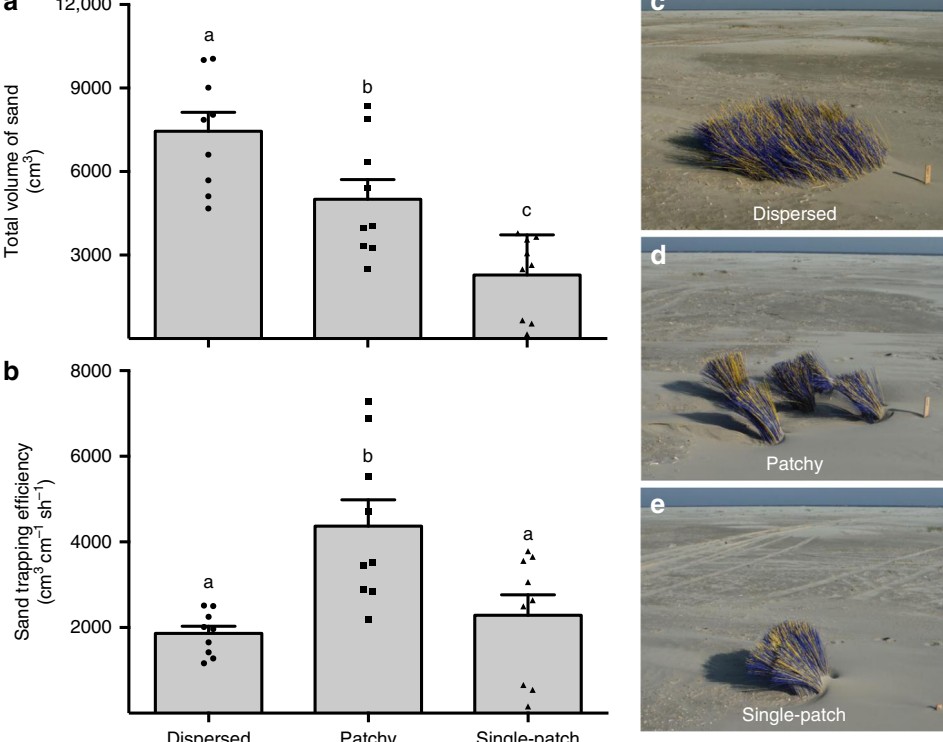

**Fig. 4** Effect of spatial organization of shoot mimics on sand capture. **a** The total trapped sand volume was highest for the dispersed and lowest for the single-patch configuration. **b** Sand-trapping efficiency (expressed as trapped sand volume divided by the distance between consecutive shoots) was more than two-fold higher in the patchy configuration. **c–e** depict the spatial mimic organization in the dispersed (Ballistic-like), patchy (Lévy-like) and single-patch (Brownian-like) configurations, respectively. Experiment was set up in three experimental blocks and measured over three different time points. Error bars represent +s.e.m. ($N = 9$, three experimental blocks, three repeated measures). Letters depict post hoc grouping ($p < 0.05$). Source data are provided as a Source Data file

Our findings reveal the existence of Lévy-like movement in plants. Although Lévy walk patterns have been found in a wide range of scientific fields, including physics, chemistry and economics, their biological application centred around explaining movement patterns of mobile lifeforms as a search optimization process for food or safety. Our results move beyond this paradigm in highlighting that (i) heavy-tailed individual-scale movement strategies underlie the formation of interconnected belowground rhizomal networks in beach grasses, and that (ii) the resulting spatial organization of aboveground shoots affects their biophysical feedback strength, thereby exerting early developmental stage control on their landscape-modifying abilities. In doing so, this study provides proof of concept for a much broader application of heavy-tailed random walks in biology. First of all, as many biogeomorphic landscapes are formed by plants[13,17–19], we suggest that heavy-tailed expansion strategies are likely not limited to beach grasses, but may also occur in for example seagrasses meadows, salt marshes and freshwater wetlands. Secondly, as networks and connectivity are fundamental to many biological processes[33], including vascular networks[34], neural brain networks[35], fungal networks[36], and the structure of insect nest networks[37], the potential of heavy-tailed random walks to explain biological network formation may well exceed this work's scope.

A mechanistic understanding of how clonally expanding plants control landscape formation may be translated into new tailor-made planting designs for the restoration of rapidly degrading biogeomorphic landscapes[19,38], or the construction of novel nature-based flood defences[39,40]. Currently, costs of creating such feedback-controlled biogeomorphic ecosystems are 10–400 times higher compared with ecosystems without strong feedbacks, and with low chances of success[41,42]. Recent work emphasizes that common designs insufficiently consider intraspecific facilitation, and suggests to clump outplants into aggregations[19]. In demonstrating that clonal grasses balance a trade-off between engineering and expansion, our work highlights the potential to optimize biogeomorphic landscape construction by creating patches large enough to generate sufficient self-facilitation, while remaining as small as possible to maximize clonal outgrowth.

## Methods

**Characteristics of clonal plant movement.** Clonal plants are able to spread laterally by producing rhizomes or stolons. Contrary to animals, clonal plants can occupy multiple places at once, with immobile shoots located on an expanding network of rhizomes[43]. By excavating the plant's rhizome/stolon network we can describe its expansion strategy by measuring only a few parameters: the branching angle (reorientation angle between successive shoot locations), the branching degree (number of shoots connected to a single shoot) and the step size (the distance between two connected shoots).

**Field survey.** We conducted a field survey on both sides of the Atlantic coast to investigate the spatial organization and expansion strategy of the two dune grass species used in this study. European marram grass (*Ammophila arenaria*) was sampled on the West Frysian barrier island of Schiermonnikoog, the Netherlands (53°30′25.38″N, 6°18′52.52″E) from April to June 2017. American beachgrass (*Ammophila breviligulata*) was sampled on two barrier islands on the east coast of the United States: Hatteras island, North Carolina (35°13′58.67″N, 75°36′6.60″W) and Chincoteague island, Virginia (38°0′25.81″N, 75°15′36.64″W) in August 2017. For both species we selected young isolated colonizing plants near the foot of the primary dunes. Earlier studies demonstrated similar tillering rates (the rate at which new shoots emerge) between species during colonization and we therefore assumed no age differences between species[26]. By cutting off the aboveground biomass and replacing each shoot by a labelled coloured pin, we were able to extract the spatial coordinates of all shoots (in cm) using a custom-made Matlab tool (see Supplementary Figure 1 for a visual description of the methods and ref. [60] for a stepwise protocol). In addition, we collected soil (at ~5 cm depth in middle of clonal individual) and leaf samples (pooled per clonal individual) to assess nutrient availability. For soil samples, % organic matter was estimated as weight loss by ignition at 550 °C. Plant available phosphorus (Olsen-P) was estimated using a bicarbonate extract analysed using an inductively coupled plasma (ICP) spectrophotometer (iCAP 6000; Thermo Fisher Scientific). Soil nitrogen percentage was determined by an elemental analyser (Carlo Erba NA1500, Thermo Fisher Scientific). After drying at 60 °C to constant weight, we grinded the leaf samples

using a ball mill (MM400, Retch, Haan). Subsequently, C and N concentrations were determined by an elemental analyser (Carlo Erba NA1500, Thermo Fisher Scientific). Lastly, leaf P concentrations were determined through digestion of 4 cm of $HNO_3$ (65%) and 1 ml of $H_2O_2$ (30%) in a microwave oven, after which the samples were diluted and analysed using an inductively coupled plasma emission (ICP) spectrophotometer (iCAP 6000; Thermo Fisher Scientific).

**Characteristics of spatial shoot organization.** The spatial shoot coordinates were extracted from still images ($N = 8$ for both species), and subsequently used for analyses on spatial clustering and complexity of the shoot organization. Using Ripley's K we tested whether the patterns differed significantly from a random homogeneous distribution. Using the normalized L-function: $L(r) = \sqrt{K(r)/\pi}$, with $r$ being the distance and $K(r)$ being Ripley's K function, we identified whether the shoots were spatially clustered (for $L(r) > r$) or dispersed ($L(r) < r$) (Supplementary Fig. 2b, f). All Ripley's K analyses were performed using the spatstat package for R[44]. Furthermore, as many self-organizing phenomena in nature show self-similarity, we calculated the fractal dimension, a complexity index, of the spatial patterns using box counting (Supplementary Fig. 2c, g). Box counting is one of the simplest methods to measure the fractal dimension: $Df = \log N(s)/\log s$, with $N(s)$ being the number of boxes of a certain size s. Hence, the fractal dimension is given by the slope of $N(s)$ on log–log scale. A pattern is fractal (self-similar) if $N(s)$ has a slope that is approximately constant, with a corresponding fractal dimension ($N(s) \sim s^{-\alpha}$) (Supplementary Fig. 2d, h).

**Measuring step sizes in the clonal network.** To identify the clonal expansion strategy of both species, we first excavated plants from our plots ($N = 9$ with 244 steps for *A. breviligulata* and $N = 5$ with 533 steps for *A. arenaria*) and noted the rhizomal connections between shoots. Using this information, we were able to manually record the step sizes, branching angles and degree using ImageJ v2.0.0 (Supplementary Fig. 1).

Next, we tested the use of two simple connection algorithms on the spatial coordinates of the shoots of individual clonal plants to see whether these methods would approximate the manually obtained step size distribution equally well thus allowing a more automated expansion of our data set. The first connecting algorithm tested was based on the Travelling Salesman principle (TS). The Travelling Salesman in a classical NP (non-deterministic polynomial time) hard problem from computer science that deals with computing the shortest possible route given N number of cities in which every city has to be visited once[45]. In our case we used a numerical approach that connected all shoots N in the clonal network in an open circuit until the total route length did not shorten anymore for N times. The second algorithm searches for the nearest neighbour (here NN) consecutively until all shoots N are connected. The algorithm was iterated N times, starting at every individual shoot, and the route with the shortest total length was selected. For both cases, the selected route was then used to describe the step-size distribution of the plants. The methods were validated using a two-sample Kolmogorov-Smirnov test (KS-test) that allows for comparing the estimated distribution against the distribution obtained from excavation. For both species, based on the KS-test, the step size distributions we obtained using both connecting algorithms were statistically indistinguishable from the ones measured in the field (*A. arenaria* (533 step sizes): TS ($p = 0.110$) and NN ($p = 0.059$); *A. breviligulata* (244 step size): TS ($p = 0.133$) and NN ($p = 0.052$)) (Supplementary Fig. 4). For the characterization of the step size distribution of both the pooled data set and the individual plants we chose the nearest neighbour (NN) method, which is simplest and holds fewest assumptions.

**Characterizing step size distribution.** We evaluated the adequacy of five commonly used candidate models to describe the observed step size distribution of both the pooled data and the individual plants. The five models correspond to the random movement strategies most used in literature. Brownian walk was used as a null-model, while Lévy, truncated Lévy, log-normal and Composite Brownian walks were compared as alternative heavy-tailed models. Maximum likelihood estimates were used for the parameters values of the models[46]. Instead of the commonly used approach for estimating the minimum step size for power laws as described in Clauset and co-authors[47], we adopted a fixed minimum step size, as we aimed to identify the distribution function that best fits the majority of our data, rather than identifying power-law behaviour in the tail. To account for the methodological measurement error, calculated from translating pixels to cm (~0.34 cm), we set the minimum step size at twice the error (0.68 cm), as it was not possible to accurately distinguish separate shoots below this minimum value.

The Brownian walk has long been the default random walk model in physics and biology and corresponds to normal diffusion. The step sizes are drawn from an exponential distribution:

$$f(s) = \lambda e^{\lambda(smin-s)} \tag{1}$$

with $s$ being the step size and $smin$ the minimum step size of the distribution. Parameter $\lambda$ was derived from the data using the maximum likelihood estimator:

$$\hat{\lambda} = \frac{n}{\sum_{i=1}^{n}(s_i - smin)} \tag{2}$$

where $n$ is the number of shoots.

The Lévy walk is a model for describing movement that corresponds to anomalous diffusion. Its scale-free properties are modelled with a Pareto distribution, which follows a power law:

$$f(s) = \left((\mu - 1)s\mathrm{min}^{\mu-1}\right)s^{-\mu} \qquad (3)$$

with $s$ being the step size and $s\mathrm{min}$ the minimum step size of the distribution. The Lévy or scaling exponent $\mu$ determines the shape of the distribution. When $1 < \mu < 3$, the movement is referred to as a Lévy walk. However, when $\mu$ is very close to 1, the movement becomes ballistic as the probability of making very large steps increases. As $\mu$ approaches 3, it approximates a Brownian walk (and becomes truly Gaussian at $\mu > 3$). Parameter $\mu$ was derived from the data using the maximum likelihood estimator[46,48]:

$$\hat{\mu} = 1 + \frac{n}{\sum_{i=1}^{n}(\ln(s_i) - \ln(s\mathrm{min}))} \qquad (4)$$

In biology, scale-free properties are confined to a certain spatial range by physical constraints and some people refrain from fitting unbounded Pareto distributions on their data[49]. Nevertheless, the majority of studies on Lévy walk behaviour do include probability distributions with unbounded means to describe their empirical data in addition to bounded distributions[1,4,46,50]. This is because, although the unbounded Pareto distribution does not include an upper bound (maximum step size), it may provide an accurate enough description of the empirical data when the maximum step size is beyond the scale of measurements (in our case: $s\mathrm{max} > 1.4$ m).

A more commonly observed distribution in nature is a truncated Lévy. A truncated Pareto distribution has a maximum step size and therefore expresses exponential decay in the tail of the distribution. The probability density function for a truncated Pareto distribution is given by:

$$f(s) = \frac{\mu - 1}{s\mathrm{min}^{1-\mu} - s\mathrm{max}^{1-\mu}} s^{-\mu} \qquad (5)$$

with $s\mathrm{max}$ being the maximum measured step size. The maximum likelihood estimate for $\mu$ can be determined numerically using the log-likelihood given in Eq. (6) to find the $\mu$ that satisfies $\frac{dl}{d\mu} = 0$.

$$l = n \ln\left(\frac{\mu - 1}{s\mathrm{min}^{\mu-1} - s\mathrm{max}^{\mu-1}}\right) - \mu \sum_{i=1}^{n} \ln s_i \qquad (6)$$

There are other candidate models that are heavy-tailed and roughly follow a straight line on a log–log scale and their cumulative distribution functions are therefore difficult to distinguish from pure power laws. Most commonly used are hybrid exponential and log-normal distributions[47,51,52]. The probability density function $g(s)$ of a two-component Brownian model is given by:

$$g(s) = w_1 f_1(s) + (1 - w_1)f_2(s) \qquad (7)$$

where $f_1(s)$ and $f_2(s)$ are two exponential distribution functions as described in Eq. (1) and $w_1$ the relative weight between the two functions. The maximum likelihood estimate of the parameters was obtained by maximizing the log-likelihood function $L = \sum_{i=1}^{n} \log(g(s))$ numerically.

Finally, we considered the log-normal probability density function:

$$f(s) = \frac{1}{2\sigma\sqrt{2\pi}} \exp\left[\frac{-(\ln s - \mu)^2}{2\sigma^2}\right] \qquad (8)$$

with maximum likelihood estimates of $\mu$ and $\sigma$ being the mean and standard deviation of the log-transformed data, respectively.

Model selection was based on the weighted Akaike information criteria (AIC), which allows for comparing the relative differences between models[53].

$$w\mathrm{AIC}_i = \frac{\exp(-0.5(AIC_i - AIC_{\min}))}{\sum_{k}^{n} \exp(-0.5(AIC_k - AIC_{\min}))} \qquad (9)$$

where $n$ indicates the number of models tested and the AIC values are calculated using their associated log-likelihood and the number of parameters estimated[46]. Following the methods proposed by Clauset and co-authors[47], we tested the goodness of fit of the candidate models (Lévy, truncated Lévy, composite Brownian and log normal) using one-sample Kolmogorov-Smirnov tests (Supplementary Fig. 4).

We used these methods to select the best models for the total data set of 752 step sizes for *A. breviligulata* (consisting of step size data from 18 individual clonal plants) and 1471 step sizes for *A. arenaria* (consisting of step size data from 17 individual clonal plants). In addition, we described the individual movement pattern for 12 *A. arenaria* and 4 *A. breviligulata* individuals for which sufficient data ($n > 30$) were available (see Supplementary Table 1 and Supplementary Fig. 6). We consistently found (truncated) Lévy or Composite Brownian to best describe our data, regardless of the number of shoots in the network with a $T\mu = 1.96 \pm 0.06$ (mean $\pm$ s.e.m.) for *A. arenaria* and a $T\mu = 1.54 \pm 0.05$ for *A. breviligulata*. We therefore assume that the clonal expansion strategy of beach grasses is stationary during early dune development.

**Random walk model.** We coupled a random walk simulation model to a biophysical model to investigate how the expansion strategy of establishing plants affects sand capture. To obtain empirically accurate results, we first tested the

complexity of the random walk model required to adequately capture the clonal expansion behaviour of both dune grasses. To this end, we compared the spatial pattern characteristics (using the fractal dimension (Df) of the generated pattern) of a default, random walk model with more complex models and our empirical data (see Supplementary Fig. 2). Specifically, we tested to what extent the simplest, one-directional (i.e. non-branching) model could be improved by including an algorithms that allow for (1) branching and/or (2) a correlated turning angle derived from our field data (Supplementary Fig. 3). Hence, we simulated the following four different model combinations: (1) branching + random angle, (2) branching + correlated angle, (3) no-branching + random angle, and (4) no-branching + correlated angle, and compared the results over the range of scales used in our empirical data (2 ~ 16 cm) with a linear mixed effect model using angle and branching as fixed factors and model run as random effect. We found no significant effects of either branching ($F_{1,103} = 0.03$; $p = 0.870$, $N = 7$) or turning angle ($F_{1,103} = 0.14$; $p = 0.709$, $N = 7$) on the fractal dimension of the pattern (over the range of 2 ~ 16 cm). Furthermore, we found no significant differences in fractal dimension of our model-generated patterns compared with our field data (model: Df ~ 0.73 and field: Df ~ 0.77; $t_{80} = 1.38$; $p = 0.171$, $N = 7$ for both field and model). We therefore used the simple, default random walk in our further analyses.

**Spatially explicit biophysical model.** We explored the effect of differences in clonal expansion strategies (as expressed by their step size distribution) on the potential of an individual clonal plant to capture sand with the use of a spatially explicit model in an infinite domain. As our aim was to merely examine the effect of shoot organization on wind flow as a proxy for sand capture potential, we constructed a simple model that disregarded many aspects of the complex phenomenon of natural dune formation. In this minimal model, we assume a constant unidirectional flow, no initial beach topography, differences in grain size distribution nor sand moisture, which are all known to affect transport threshold and shear stress at the sand surface[54]. Furthermore, we simulated the spatial organization of shoots as the result of a discrete simple random walk (see previous paragraph for the validation of the random walk model), taking random step sizes from a truncated Pareto distribution with a Lévy exponent ranging from: $1 < \mu \le 3$.

$$S(X) = \left(X\left(s\mathrm{min}^{1-\mu} - s\mathrm{max}^{1-\mu}\right) + s\mathrm{max}^{1-\mu}\right)^{1/(1-\mu)} \qquad (10)$$

where $X$ is a random uniformly distributed variable ($0 \le X \le 1$), $s\mathrm{min}$ the minimum step size (set at the minimum step size of our field data: 0.34 cm) and $s\mathrm{max}$ the maximum step size (set at the maximum step size: 75.33 cm from our field data).

After a simulation was finished, we modelled the effect of the shoot organization on potential area of sand deposition by applying a convolution matrix with the effect of a single shoot on the incoming wind flow to all shoots on the spatial grid. The convolution matrix was constructed by simulating the wind as a unidirectional laminar flow with the viscosity of air around a single shoot ($\varnothing$ 1.5 mm[23]) (with the use of the 2D computational fluid dynamics (CFD) software of ANSYS R17.2 (ANSYS® CFD™[55]). As the effect of plant morphology on sand capture is greatly mitigated by shoot density[23], we assumed a simple plant geometry in our model and shoot basal area alone was used to characterize the interaction between vegetation and wind flow[54]. The incoming wind speed was set to 6.5 m s$^{-1}$, which corresponds to the average wind speed along the coast of the Dutch Wadden Sea Islands[56]. The resulting changes in wind speed were translated to the potential area of sand deposition by calculating the sedimentation threshold as a proportion (~61%) of the incoming wind speed based on the results from Davidson-Arnott and Bauer[57].

Using the discrete random walk approach, we simulated differences in shoot organization for a given number of shoots. Next, we calculated the sand-trapping efficiency by dividing the total area of sand deposition by the average inter-shoot distance. Simulations were run for a range of Lévy exponents ($1 < \mu \le 3$) and a varying number of shoots (Supplementary Fig. 7). Both sand deposition and trapping efficiency were plotted as a function of $\mu$. To test the robustness of our results, we calculated the Lévy optimum (ranging from $\mu = 1.5$ to 3.0 with increasing steps of 0.5) for a range of critical thresholds for sand deposition (0.25–0.85%) and an increasing number of shoots (minimum 30, maximum 5000 shoots). The strategy yielding the highest potential area of sand deposition or sand-trapping efficiency was determined by comparing the mean $\pm$ s.e.m. for the different strategies (Supplementary Table 3). To validate the use of a simplified laminar flow in our biophysical model, we compared the outcome of our field experiment (see next section) with simulated shoot patterns that reflect the shoot organizations we used in our experiment. We found the results to be consistent, that is, potential sand deposition was highest in the more dispersed shoot organization whereas sand-trapping efficiency was highest in the patchy organization (Supplementary Fig. 8).

The model was implemented in MATLAB version R2015b (©1984-2016, The Mathworks, Inc.).

**Field experiment.** We conducted a field experiment on a bare beach plain of Schiermonnikoog, the Netherlands (53°30′36.73″N, 6°19′37.84″E) in the summer of 2016, to test the effect of the spatial shoot organization on the sand-trapping ability. We constructed plots of 4 m$^2$ in which we placed flexible artificial dune grass mimics (three plastic bristles, diameter 0.2 cm, length 75 cm, made up one shoot[23]) in three spatially distinct patterns (dispersed, patchy and single-patch).

In total ~2000 bristles were inserted in 4 m² PVC templates (which resulted in 500 shoots m⁻¹, a natural shoot density[31] previously used in biophysical studies[23]) with the spatial patterns drilled into them and attached to wooden beams in 20 cm deep pits on the beach, after which we refilled the plots using drift-sand resulting in a canopy-height of the mimics of 55 cm. Each treatment was replicated three times in a randomized block design that also included a control plot (only PVC sheet, no bristles) per block, yielding 12 plots in total.

Sand deposition was measured every month (June, July and August) on a 0.1 × 0.1 m scale with the use of a sediment erosion bar construction. We determined the total volume of sand capture by calculating the amount of sand on each plot, corrected by the overall block-level change in bed level obtained from the control plots. The sand-trapping efficiency was calculated by dividing the volume of sand by the average inter-shoot distance.

We used a linear mixed-effects model with a Satterthwaite approximation of the degrees of freedom to test the effect of the spatial organization on both sand deposition and sand-trapping efficiency, using time of measurement and block as random effects. Tukey HSD posthoc tests were used to separate treatment effects.

**Reporting summary**. Further information on research design is available in the Nature Research Reporting Summary linked to this article.

## Data availability
The plant still images, shoot coordinates, step length data, experimental data and model results that support the main findings of this study are available via the Data Archiving and Networked Services (DANS) EASY (https://doi.org/10.17026/dans-z45-kc6k)[59]. In addition, the source data of Figs. 2–4 and Supplementary Figs. 2–4, 6–8 and Supplementary Table 3 are provided as a Source Data file. All other relevant data is available upon request.

## Code availability
The code (developed in Matlab R2015b) used in the survey to extract step sizes from images and the Matlab scripts for running the biophysical model can be accessed via the Data Archiving and Networked Services (DANS) EASY (https://doi.org/10.17026/dans-z45-kc6k). A full protocol for extracting step sizes from clonal plants in the field can be found at the Nature Research protocol exchange (https://doi.org/10.21203/rs.2.9545/v1)[60].

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

## Acknowledgements

We thank H. de Kroon, B. Silliman, R. Oldenkamp and F. Bartumeus for their valuable comments on previous versions of our paper. We thank many students, volunteers, technical assistants and especially L. Hendriks, S. Wössner and H. Wiersema for their help during the setup and measurements of the field experiment. We thank Natuurmonumenten for their permission to conduct the experiment and to perform field measurements at the National park Schiermonnikoog. We thank K. Holcomb and B. Harrison from Chincoteague National Wildlife Refuge and Alligator River & Pea Island National Wildlife Refuges, respectively, for their help and permission to perform measurements on Hatteras and Chincoteague island. V.C.R. was financially supported by NWO-Building with Nature grant 850.13.052. K.S. was supported by the National Key R&D Program of China (2017YFC0506001), the National Natural Science Foundation of China (41676084) and the EU Horizon 2020 project MERCES (689518). The work of J.v.B is funded by the VNSC project "Vegetation modelling HPP" (contract 3109 1805). L.L.G. was supported by NWO-Veni grant 016.Veni.181.087. T.v.d.H. was supported by NWO-Vidi grant 16588.

## Author contributions

V.C.R., T.v.d.H. and J.v.d.K. conceived the idea. L.P.M.L., K.S., J.v.B. and T.J.B. helped further conceptualizing the idea. S.H., V.C.R., A.C.W.B. and L.L.G. collected the field data. K.S., J.v.B., S.H. and V.C.R. conceptualized and constructed the biophysical model. J.H., V.C.R. A.C.W. and T.v.d.H. conceptualized and conducted the field experiment. S.H., V.C.R. and K.S. performed all data analyses. V.C.R. and T.v.d.H. wrote the first draft of the paper and all authors contributed to the subsequent drafts.

## Additional information

**Peer Review Information:** *Nature Communications* thanks Orencio Duran Vinent and other anonymous reviewer(s) for their contribution to the peer review of this work. Peer reviewer reports are available.

