## [Peer Review File · Nature Communications]

Reviewers' comments:

Reviewer #1 (Remarks to the Author):

The paper "A Lévy expansion strategy optimizes dune building by beach grasses" seeks to describe the self-organization of coastal dune grasses and how they affect or influence geomorphology (i.e. dune height) through sand trapping efficiency using field measurements from two *Ammophila* species (from Germany and US), a spatially-explicit model, and a field experiment manipulating density of artificial "grasses". The authors hypothesize that these dune grasses employ a Lévy-type expansion strategy of "multiple dense shoot patches that maximize self-promoting feedbacks at the landscape scale with a minimum investment in their belowground rhizome network".

The topic is timely, however, there are issues with the claims made by the authors as well as recent and highly relevant literature that has not been included. The most important of these would be exclusion of a recent paper by Zarnetske et al. 2015 (*Journal of the Royal Society*), which comes to the same conclusion made by the authors regarding the differences in both *Ammophila* species' growth, ability to capture sand based on growth strategy, and how this affects dune development with variable rates of sediment supply. They conclude "High sand supply also promotes a positive feedback with *A. breviligulata* by stimulating more horizontal growth, which reinforces its lower tiller density and lower sand capture efficiency [18,24,58]. Sand capture efficiency is likely to be the most important factor that distinguishes how these species shape dunes...". The current paper findings are intriguing and complementary to this work, however, the way the paper is written inaccurately represents the literature and makes claims to novelty that have already been published without proper acknowledgment.

In lieu of this most significant oversight, I have other comments regarding the paper that can help to improve the contribution of the work and put it into the context of dune-biogeomorphic feedbacks. The motivation for the work appears to be focused on how plants build dunes to the height and extent that they do, however the timescale of their study and lack of discussion on other important factors in dune development that occur in most papers on this topic (i.e. sediment supply, disturbance, growth response of plants in response to sediment accumulation) are major limitations.

The reviewed literature in the introduction does not include recent works that is the state of knowledge of general lateral expansion rates and growth in dune species. Goldstein et al. 2017 (*Geomorphology*) reported how lateral vegetation growth rates exert control on coastal foredune development. Papers by Stallins (see Corenblit and Stallins, 2018, *Geomorphology*) and findings in Goldstein et al. 2017 indicate that disturbance controls dune topography, which is tied to lateral expansion rates. The same species can exhibit both low dunes and tall, long dune ridges due to slow lateral growth and rates of disturbance. The results from Zarnetske et al. 2012 are not accurately portrayed in this paper. They found that tiller density and the feedback with sediment supply result in the differences in dune morphology (rhizome internode length was also measured in the study), thus the claim on lines 82-83 is untrue.

The hypothesis assumes that lateral growth determines investment in the belowground rhizome network; however, without measurements of belowground tissue, this assumption is faulty based on the literature. The statement (lines 131-132) that equally long rhizomes result in the development of more belowground tissue relative to the photosynthesizing aboveground shoot cannot be assessed without above or belowground tissue measurements. There is sufficient literature demonstrating inter- and intraspecific differences in root:shoot ratios and biomass allocation including with sediment accumulation/burial (which would be expected in growing dunes). See the following papers for more on this topic: Brown 1997 (*Journal of Ecology*), the numerous papers reviewed in Gilbert and Ripley 2010 (*Austral Ecology* - especially papers from Table 2 that examine the *Ammophila* species of interest), Charbonneau et al. 2016 (*Journal of Environmental Management* - incredible root:shoot ratio

differences and rooting depths between species), Brown and Zinnert 2018 (Ecosphere), among many others.

Given that dune plants are known to respond both above- and belowground to sand deposition, the field design using artificial stakes and over such a short timeframe does not fully represent sand capture by plants, nor the important biogeomorphic feedback between plants and sediment. More detail is needed to understand the relevance to dune grasses. It is unclear what the density of your artificial shoots are and whether this represents densities of plants seen in the field. Again, the formation of tall dune ridges is also a function of coalescing time among many smaller dunes, thus the timescale of the study for sand trapping is difficult to extrapolate to landscape scale phenomena. The similarities between the findings of this paper and the Zarnetske et al. 2012 and 2015 papers could be used to enhance our understanding of dune growth, but this paper needs to be put into context of the recent literature.

Reviewer #2 (Remarks to the Author):

I read with great interest the manuscript. In particular, the idea that the spatial organization of clonal expansion of dune-building grasses arises from the competition between efficient growth (plant resources) and maximum sand accretion is quite attractive. However, I find the current manuscript didn't really answer the problem of optimization of dune building and essentially just describes how both grass species are spatially distributed. The main problem is that if a more dispersed clonal growth (as in *A. Brevigulata*) always leads to higher sand deposition, which is after all the central metric for dune formation, why then conclude a less spread, more patched growth strategy is better? I understand this strategy decreases the total belowground biomass but it is not obvious why this is a problem if enough nutrients are present. I think the issue here is that the authors don't include neither time as a variable nor the effect of potential sand erosion. Both are mentioned in the discussion, but not explicitly added as part of the feedback selecting a given strategy. Below are several points I think the authors should address before I can recommend publication:

Major points:

1. Role of time: time is a critical variable as we know that given enough time plants will cover most of the soil. The authors should clarify in which part of the plant colonization phase were the measurements taken. As mentioned above, they should also consider the rate of expansion between both species. I mean, if the relevant metric for dune growth is sand deposition per unit time, then how fast plants colonize a given area is as important as the size of the sand deposition area.
2. Relevance of dune height: Although there is evidence that both species leads to different dune heights, I don't see its relevance for the dune building optimization problem. If one species capture more sand, that would be more relevant for dune building than the fact final dune height is lower.
3. Relevance of sand trapping efficiency: I don't see why this metric is relevant compared with total sand accretion area. The authors should explain better why total belowground biomass can hinder clonal expansion and thus give a competitive advantage to *A. Arenaria*.
4. Universality of $\mu=2$: Although the introduction of sand trapping efficiency is interesting, the maximum efficiency at $\mu=2$ is completely dependent on the wind and sand accretion models (which contains very strong assumptions and it is quite unrealistic) and there is no reason why that value should be universal. The authors should highlight this fact and modify the conclusions as necessary.

5. Wind and sand accretion models: The assumptions for the model are quite unrealistic, as the authors use a laminar 2D flow over what is seems like a single rigid cylinder (simulating a single plant shoot), instead of the real 3D turbulent flow over a complex and flexible plant geometry. Also the threshold for sand transport is not really function of wind velocity (which changes with elevation) but wind shear stress within the turbulent boundary layer. The authors should address the limitations of the model and discuss the dependency of sand deposition area on plant distribution in more general terms. For instance, assuming each shoot has a well defined sheltered area where sand transport is negligible, the results on supplementary figure 6 can be understood as a decrease of total sheltered area (which increases linearly with the number of shoots) with the overlap between shoots. This overlap is minimum when shoots are spread (highest deposition area, low μ) and maximizes for a brownian random walk (lowest deposition area, $\mu \rightarrow 3$).

6. Fractal dimension: I don't think it is correct to even define a fractal dimension if the power law scaling is over a range of less than one decade as in supplementary figure 2. In the absence of such a range in the data, the authors should not really invoke any scale invariant or fractal arguments in their results or discussion, unless there are process-based arguments to argue for a scale invariance in plant colonization or clonal expansion.

Minor points:

1. Relation to plant physiology: related to the previous point, there is no discussion of plant physiology even if we can think that ultimately clonal expansion (as related to the spatial and temporal frequency of shoot production) is a product of it. The authors could at least mention some underlying processes behind such strategies and also explore potential reasons why optimal sand accretion could feedback into a particular clonal expansion strategy. After all, the authors seem to suggest the growth of *A. Arenaria* has evolved to maximize the efficiency of sand deposition.

2. Pareto distribution (Eq. 3): there seems to be a problem with the definition of the distribution as the average value of the step size using Eq.3 diverges for $\mu=2$ instead of $\mu=1$. I think the power in Eq.3 should be $\mu+1$. The authors should confirm this is just typo in the equations.

In summary, in the current version, the authors don't provide sufficient evidence to back one of their main conclusion. This should be addressed before the manuscript can be accepted for publication.

Orencio Duran

Reviewer #3 (Remarks to the Author):

I. Overview

The authors addressed a very relevant problem – the formation of biogeomorphic landscapes – and connected it to how specific types of plants expand in the space. Their striking result is an observation of a Levy-like movement being performed by those plants, and they connect it to the structure of the sand dune that it forms. I found the reported results very appealing to the Levy flight community, because of the new functional role of this type of strategy. To my knowledge, the results are original. However, I think the work would benefit from reviewing some technical aspects of the analysis and modeling that I judge relevant and will describe below.

II. General comments

1. Stationary condition. When the plants were selected to be analyzed, how one can infer if the spatial pattern is already stationary? If patterns of different ages were compared, I wonder if one can get heterogeneities that contribute to the Levy pattern statistics due to this effect.
2. Box counting. I did not find any reference in the main text to the results of the fractal dimension analysis. Is this analysis relevant to the results, or the Ripley's K function is sufficient for their claims of heterogeneous spatial distribution?
3. Typo. There is a typo in Equation (1), the pdf of the exponential distribution. It should be:
 $f(s) = \lambda e^{-\lambda(s - s_{\min})}$
4. Equation (6). From where this expression for estimating the exponent of a TP distribution comes from? It is different from the one that I am aware of (White et al. Ecology 89 (2008)).
5. Typo, line 310. The probability density function $g(s)$... The notation used in equation (7) is $f(s)$.
6. Equation (7). It would be good if the authors could be clearer about how they estimated the parameters for a composite Brownian model. In particular, since you are adding more parameters in this fitting (two exponents plus one weight), how do you penalize your MLE?
7. Equation (8). I did not see the statistics of a lognormal distribution in the table or in the text. I think it should be included in the table, because it is always a *strong* alternative to a Levy model.
8. Model. About the spatially explicit model: (i) What type of boundary condition (if any) was used in building the spatial location of the steps? (ii) What are the sizes of the maximum step and environment? (The authors only report the minimum size as 0.34 cm..) These parameters are extremely important to interpret the model.
9. Levy dust. There is a key word for the spatial locations visited by a Levy walker (what is plotted in Figure 3); this set of points is called a Levy dust. I suggest the authors to use this key word in their paper because it will draw attention to it.
10. Figure 3. It would be very good if the authors included a scale bar so we can infer the distances in figures a-c.
11. "However, when μ approaches 1, the movement becomes ballistic as the probabilities of making very large steps or smaller steps become equal". I find this sentence confusing, and I did a numerical experiment to check it, because I was curious. I generated power-law distributed random numbers with different exponents and checked for the relative orders of magnitude of my samples. I did not get the same proportion of small and large values if $\mu \rightarrow 1$ (I got much more larger values). For the two references cited that I have access (de Jager and Bartumeus), I did not find support for this claim.

III. Comments regarding the analysis

1. Why the authors did not include in their fitting of statistical models a fit of the minimal step? Since the most important part of identifying Levy flights is the tail of the distribution, I wonder why they decided to apparently fix the minimal step in their analysis. The cited reference (Clauset) has a discussion on why it is necessary to fit the minimal step, and I think the statistical power of the results would benefit from a discussion about the reasons of not including a minimal step fit, or even doing it.
2. How many synthetic distributions were generated for the model comparison? I could not find this information in the paper.
3. Figure 2. I do not find this cumulative plot very informative. It would be nice to include, for instance, the best fit Levy, composite, and lognormal distributions for both data sets. Maybe split this into two figures, one per type of plant.
4. One of the main properties of Levy flights statistics is the combination of very long steps and short steps. I suggest the authors to include the information of both the minimum and maximum shoots measured in their statistics table 1, for each plant included in their analysis.
5. Supp. Fig. 5: The data shows individual plants in different colors and the combination of all the data

in thick markers. In the case of *A. arenaria*, it is evident that the tail statistics (steps larger than 10 cm) is dominated by only a few individuals. Therefore, I think it can be a caveat in the global analysis if the authors simply combine different individuals in the same dataset, because it can be the case that the heterogeneities in the plants are contributing to the power law statistics.

Overall conclusion:

As I was reading the paper, I noticed several technical problems on the statistical analysis and missing information about how it was done. Of particular concern is the lack of details in table 1, and also the lack of alternate distribution functions that could be tested (e.g., the lognormal one). I find the works by Humphries et al. particularly good references of how to treat the data in the case of Levy flights (see their Methods in Ecology and Evolution 2013 paper). I understand the technical difficulties of, say, acquiring more data to improve the statistics, but I think that to argue for a Levy-type statistics one has to do an investigation that is deeper than what is currently presented in the paper. I hope my comments will be useful in this direction.

Reviewer #1 (Remarks to the Author):

- 1) The paper “A Lévy expansion strategy optimizes dune building by beach grasses” seeks to describe the self-organization of coastal dune grasses and how they affect or influence geomorphology (i.e. dune height) through sand trapping efficiency using field measurements from two *Ammophila* species (from Germany and US), a spatially-explicit model, and a field experiment manipulating density of artificial “grasses”. The authors hypothesize that these dune grasses employ a Lévy-type expansion strategy of “multiple dense shoot patches that maximize self-promoting feedbacks at the landscape scale with a minimum investment in their belowground rhizome network”.

The topic is timely, however, there are issues with the claims made by the authors as well as recent and highly relevant literature that has not been included. The most important of these would be exclusion of a recent paper by Zarnetske et al. 2015 (Journal of the Royal Society), which comes to the same conclusion made by the authors regarding the differences in both *Ammophila* species’ growth, ability to capture sand based on growth strategy, and how this affects dune development with variable rates of sediment supply. They conclude “High sand supply also promotes a positive feedback with *A. breviligulata* by stimulating more horizontal growth, which reinforces its lower tiller density and lower sand capture efficiency [18,24,58]. Sand capture efficiency is likely to be the most important factor that distinguishes how these species shape dunes...”. The current paper findings are intriguing and complementary to this work, however, the way the paper is written inaccurately represents the literature and makes claims to novelty that have already been published without proper acknowledgment.

Reply: Thank you for reviewing our paper. We agree that we could have cited the suggested paper (Zarnetske et al. *J. R. Soc. Interface*, 2015) and a few others as well to more precisely portray the context of our findings. Indeed, previous studies combined conclude that species-specific shoot densities affect sand capture and vice versa, and in turn relate this to observed differences in dune morphology. However, to our knowledge, there currently exists no mechanistic explanation for how colonizing beach grasses spatially organize their shoots (i.e. given a certain number of shoots) to optimize their sand trapping potential. As such, our paper investigates spatial heterogeneity on an individual plant level, rather than analyzing mean values on a species level. Therefore, our paper goes beyond (but builds on) earlier work by demonstrating that beach grasses employ Lévy-like random walk strategies – previously only observed in mobile organisms – to rapidly initiate dune formation and escape physical stress. As this is the first time that such strategies are described for clonally expanding plants, the relevance of these findings are most likely not limited to coastal dunes, but also applicable in other biogeomorphic landscapes.

We have now modified the text to more accurately include previous work on the interaction between dune formation and plant growth:

p. 4 L83-L88: *“In addition, the plants differ in their physiological tolerance to burial and flooding stress, respectively, with *A. arenaria* being more resistant to burial stress by developing vertically expanding rhizomes, while *A. breviligulata* has a higher salinity tolerance. This suggests that both species have adopted different dune-building strategies to cope with the stressful conditions of growing at the land-sea interface²⁶. So far, studies on the biophysical feedback strength of the two species have related observed differences in dune morphology to species-specific differences in shoot densities in existing dune fields and their growth response to sand burial. Specifically, they conclude that (i) higher shoot densities promote sand capture with *A. arenaria* typically generating more shoots per square meter than *A. breviligulata* in existing dune fields, and (ii) the shooting rate of *A. arenaria* is stronger stimulated by sand capture*

compared to A. breviligulata. Yet, it remains to be elucidated whether dune-building grasses control biophysical engineering strength via the spatial arrangement of their shoots in the beach colonization phase when initiating dune formation is vital for escaping physical stress from flooding.”

Furthermore, to emphasize the novelty of the discovered Lévy-strategy for not only dune-building, but landscape-forming plants in general we have rephrased the following sentences:

p.9 L198-202: *“Overall, our work builds on previous studies suggesting that differential growth strategies can help explain the emergence of contrasting dune morphologies^{23,31,32}, by demonstrating that beach grasses adopt distinct colonization strategies that determine their engineering strength in these early developmental stages.”*

p.10 L210-219: *“Our results move beyond this paradigm in highlighting that (i) heavy-tailed individual-scale movement strategies underlie the formation of interconnected belowground rhizomal networks in beach grasses, and that (ii) the resulting spatial organization of aboveground shoots affects their biophysical feedback strength, thereby exerting early developmental stage control on their landscape-modifying abilities. In doing so, this study provides proof of concept for a much broader application of heavy-tailed random walks in biology. First of all, as many biogeomorphic landscapes are formed by plants^{13,17-19}, we suggest that heavy-tailed expansion strategies are likely not limited to beach grasses, but may also occur in for example seagrasses meadows, salt marshes and freshwater wetlands.”*

- 2) In lieu of this most significant oversight, I have other comments regarding the paper that can help to improve the contribution of the work and put it into the context of dune-biogeomorphic feedbacks. The motivation for the work appears to be focused on how plants build dunes to the height and extent that they do, however the timescale of their study and lack of discussion on other important factors in dune development that occur in most papers on the this topic (i.e. sediment supply, disturbance, growth response of plants in response to sediment accumulation) are major limitations.

Reply: Although we do discuss the potential implications of the plants’ expansion strategies to the ultimate shape and height of the dunes built, this was certainly not the core motivation of our work. Instead, our aim was to investigate how species-specific shoot organizations emerge through distinct clonal expansion strategies and how this impacts engineering strength of landscape-forming species in the very early colonization phase – i.e. embryonic dune formation. To clarify this, we have now changed the title to: *“A Lévy expansion strategy optimizes early dune building by beach grasses”* and rephrased the following sentences:

p. 3 L68-71: *“Whereas the importance of both rapid colonization and the initiation of landscape-building feedbacks is now well-recognized¹³, it remains unknown if colonizing landscape-forming plants spatially organize their shoots to combine the needs for tight patch formation and clonal expansion”*

p. 4 L94-100: *“Yet, it remains to be elucidated whether dune-building grasses control biophysical engineering strength via the spatial arrangement of their shoots in the early beach colonization phase when initiating dune formation is vital for escaping physical stress from flooding. Using random search models we aim to unravel (i) whether dune-building species differ in their clonal expansion strategy and (ii) whether the observed expansion strategies and the resulting spatial shoot organizations can be related to the sand trapping potential in these early phases.”*

Furthermore, we of course agree that other factors in addition to the plants' clonal expansion strategy contribute to the formation of the overall coastal dune landscape. We now discuss other factors that impact coastal dune formation once these plants have established and formed embryonic dunes.

p.9 L202-205 “*Once these plants have successfully established, coastal dune formation is then further steered by biophysical feedbacks between sediment supply, growth response of vegetation to sediment accumulation and the rate of disturbances that negatively impact vegetation survival*^{20,25,31}.”

- 3) The reviewed literature in the introduction does not include recent works that is the state of knowledge of general lateral expansion rates and growth in dune species. Goldstein et al. 2017 (Geomorphology) reported how lateral vegetation growth rates exert control on coastal foredune development. Papers by Stallins (see Corenblit and Stallins, 2018, Geomorphology) and findings in Goldstein et al. 2017 indicate that disturbance controls dune topography, which is tied to lateral expansion rates. The same species can exhibit both low dunes and tall, long dune ridges due to slow lateral growth and rates of disturbance. The results from Zarnetske et al. 2012 are not accurately portrayed in this paper. They found that tiller density and the feedback with sediment supply result in the differences in of dune morphology (rhizome internode length was also measured in the study), thus the claim on lines 82-83 is untrue.

Reply: Although the topography of mature dunes is not the focus of our paper, we do agree that we should have more thoroughly introduced the topic of dune-biogeomorphic feedbacks (see previous comments at point 1). Indeed, Zarnetske et al. (*Ecology*, 2012) conclude that a biophysical feedback between tiller density and sand capture results in differences in dune morphology. However, they did not investigate the effect of spatial shoot organization or the underlying mechanisms, on which this paper focused. In fact, the authors remark: “*While random tiller placement in the wind tunnel allowed us to separate the effect of species from tiller density, this tiller arrangement does not necessarily reflect the natural growth form in the field.*”

Furthermore, as our work investigates the effect of spatial heterogeneity in individual plants, we have examined the step size distribution of inter-shoot distances, which is very different from the mean rhizome internode length. We have modified the text throughout the manuscript to clarify the goal of our study (see previous comments points 1 and 2).

- 4) The hypothesis assumes that lateral growth determines investment in the belowground rhizome network; however, without measurements of belowground tissue, this assumption is faulty based on the literature. The statement (lines 131-132) that equally long rhizomes result in the development of more belowground tissue relative to the photosynthesizing aboveground shoot cannot be assessed without above or belowground tissue measurements.

Reply: We believe we may have caused confusion here, as our explanation was not accurate. We agree that, for example, rhizome diameter and therefore biomass can differ depending on the involved species, and the amount of resources it gathers. However, in our paper, we examine different movement strategies, which result in different total distances that the plant needs to cover given a certain number of shoots. Therefore, the plant needs to invest more energy in strategies that yield longer overall distances. We have modified the text to explain this more clearly:

p.6 L151-153: “*The outcome changes when accounting for the relatively high energy investment*

of this dispersed strategy, which requires covering long distances relative to more clumping strategies ($\mu > 2$) (Figure 3d)."

- 5) There is sufficient literature demonstrating inter- and intraspecific differences in root:shoot ratios and biomass allocation including with sediment accumulation/burial (which would be expected in growing dunes). See the following papers for more on this topic: Brown 1997 (Journal of Ecology), the numerous papers reviewed in Gilbert and Ripley 2010 (Austral Ecology - especially papers from Table 2 that examine the *Ammophila* species of interest), Charbonneau et al. 2016 (Journal of Environmental Management –incredible root:shoot ratio differences and rooting depths between species), Brown and Zinnert 2018 (Ecosphere), among many others.

Reply: We agree that many papers looked at biomass allocation in dune grasses during sand burial. However, our paper focuses on the very early developmental phase of beach colonization (see comments points 1 and 2), when the plants have not yet trapped significant amounts of sand. The factors that influence dune formation in later stages are now included in the discussion (see comments point 2).

- 6) Given that dune plants are known to respond both above- and belowground to sand deposition, the field design using artificial stakes and over such a short timeframe does not fully represent sand capture by plants, nor the important biogeomorphic feedback between plants and sediment.

Reply: We agree that our field experiment does not answer the complete biogeomorphic feedback loop between plants and sediment. Instead, our goal was to understand how spatial shoot organization influences sand trapping potential, as this is especially important in the early stages of dune development to rapidly overcome establishment thresholds (see previous comments).

- 7) More detail is needed to understand the relevance to dune grasses. It is unclear what the density of your artificial shoots are and whether this represents densities of plants seen in the field.

Reply: We have modified the method section to include the densities used in our field experiment (500 shoots m^{-1}), which are consistent with densities observed for *A. arenaria* and *A. breviligulata* by Hacker *et al.* (*Oikos*, 2011) and those used in the wind tunnel experiment by Zarnetske *et al.* (*Ecology*, 2012).

p. 23 L494-497: *"In total ~2000 bristles were inserted in 4 m² PVC templates (which resulted in 500 shoots m⁻¹, a natural shoot density³⁰ previously used in biophysical studies²⁴) with the spatial patterns drilled into them and attached to wooden beams in 20 cm deep pits on the beach, after which we refilled the plots using drift-sand resulting in a canopy-height of the mimics of 55 cm."*

- 8) Again, the formation of tall dune ridges is also a function of coalescing time among many smaller dunes, thus the timescale of the study for sand trapping is difficult to extrapolate to landscape scale phenomena. The similarities between the findings of this paper and the Zarnetske et al. 2012 and 2015 papers could be used to enhance our understanding of dune growth, but this paper needs to be put into context of the recent literature.

Reply: We agree with the reviewer. We hope that our amendments to the text, as described above, now clarify the phase of dune biogeomorphological succession our

work is concerned with.

Reviewer #2 (Remarks to the Author):

- 9) I read with great interest the manuscript. In particular, the idea that the spatial organization of clonal expansion of dune-building grasses arises from the competition between efficient growth (plant resources) and maximum sand accretion is quite attractive. However, I find the current manuscript didn't really answer the problem of optimization of dune building and essentially just describes how both grass species are spatially distributed. The main problem is that if a more dispersed clonal growth (as in *A. Brevigulata*) always leads to higher sand deposition, which is after all the central metric for dune formation, why then conclude a less spread, more patched growth strategy is better? I understand this strategy decreases the total belowground biomass but it is not obvious why this is a problem if enough nutrients are present.

I think the issue here is that the authors don't include neither time as a variable nor the effect of potential sand erosion. Both are mentioned in the discussion, but not explicitly added as part of the feedback selecting a given strategy. Below are several points I think the authors should address before I can recommend publication:

Reply: Thank you for your helpful comments. As now more accurately described in the paper (see our replies to Reviewer 1), our aim was to investigate how species-specific shoot organizations emerge through distinct clonal expansion strategies and how this impacts sand capture in the very early colonization phase – i.e. embryonic dune formation. Indeed, results show that the more dispersed (albeit still quite patchy!) strategy traps more sand. However, we find that the more patchy strategy is more efficient in terms of the distance the plants are required to travel (and thus the energy spent doing this) to generate a certain sand trapping potential. Although more sand is generally better in this early phase, the plants need to acquire more resources for the dispersed strategy, causing a trade-off between expansion strategy and its sand trapping potential in a resource-limited environment.

We agree that we did not sufficiently explain our focus on the early colonization phase (the time component), and the relation of the strategies to resource-limitation. To solve this, we have now clarified that our paper focused on early dune formation, and included additional analyses of both soil and leaf samples, demonstrating the nutrient-limited conditions of these systems (Supplementary table 2). Please find below how we dealt with all specific comments in detail.

10) Major points:

1. Role of time: time is a critical variable as we know that given enough time plants will cover most of the soil. The authors should clarify in which part of the plant colonization phase were the measurements taken. As mentioned above, they should also consider the rate of expansion between both species. I mean, if the relevant metric for dune growth is sand deposition per unit time, then how fast plants colonize a given area is as important as the size of the sand deposition area.

Reply: We agree. To clarify this, we changed the title to "*A Lévy expansion strategy optimizes early dune building by beach grasses*" and rephrased multiple sentences throughout the text:

p. 3 L76-77: "*To test our hypothesis, we investigated how colonizing dune-building grasses*

organize their shoots to initiate dune building.”

p. 4 L94-97: “*Yet, it remains to be elucidated whether dune-building grasses control biophysical engineering strength via the spatial arrangement of their shoots in the early beach colonization phase when initiating dune formation is vital for escaping physical stress from flooding.”*

p. 10 L213-215: “*...(ii) the resulting spatial organization of aboveground shoots affects their biophysical feedback strength, thereby exerting early developmental stage control on their landscape-modifying abilities.”*

In this early developmental phase, we examined stationary shoot organization patterns to assess their sand trapping potential. We did not include growth rates in our analyses, as earlier work demonstrated the growth rate of these species (gain in tillers/shoots) to be comparable (Zarnetske et al. *Ecology*, 2012 and Baye, 1990). We have now included this assumption in the text.

p.12 L255-257: “*Earlier studies demonstrated similar tillering rates (the rate at which new shoots emerge) between species during colonization and we therefore assumed no age differences between species*²⁵.”

- 11) 2. Relevance of dune height: Although there is evidence that both species leads to different dune heights, I don't see its relevance for the dune building optimization problem. If one species capture more sand, that would be more relevant for dune building than the fact final dune height is lower.

Reply: From a coastal management perspective, higher dunes are regarded superior, as they are generally better at protecting the hinterland against storms than lower dunes that experience frequent overwash (see e.g. Seabloom et al, *Global Change Biology*, 2013). From a plant perspective, capturing sand locally and promoting a vertical raise of habitat can be beneficial – especially in the early development stages – as it decreases the flooding probability. However, it can also be disadvantageous in later development stages when excessive burial obstructs photosynthesis (e.g. Yuan et al., *Functional Ecology*, 1993; Kent et al. *Annals of Botany*, 2005). As a consequence, a colonization strategy that optimizes sand trapping within the vegetated patch – instead of distributing it over a larger area – may over time prevent sediment depletion and promote the formation of these high, but narrow coastal dunes.

We have modified the introduction and discussion to clarify that species-specific strategies appear optimized to build different coastal landscapes that reflect the physiological tolerance of the species and may therefore have a different protective value:

p. 4 L78-88: “*However, the size and shape of these dunes and thus their ability to defend the hinterland can differ greatly depending on the dune-building species involved*²⁴. For instance, *Ammophila arenaria* (European marram grass) forms tall and steep dunes, whereas dunes formed by its North American congener, *Ammophila breviligulata* (American beachgrass) are much lower and wider and therefore considered less effective in protecting the hinterland – even when growing in the same environment (Figure 1)^{24,25}. In addition, the plants differ in their physiological tolerance to burial and flooding stress, respectively, with *A. arenaria* being more resistant to burial stress by developing vertically expanding rhizomes, while *A. breviligulata* has a higher salinity tolerance. This suggests that both species have adopted different dune-building strategies to cope with the stressful conditions of growing at the land-sea interface²⁶.”

p. 9 L190-198: “*Specifically, we found that the Lévy-like strategy of *A. arenaria* maximizes sand*

trapping efficiency by accreting sediment within multiple shoot patches, while the more dispersed strategy of A. breviligulata maximizes total sand capture over a wider area. Previous studies found that, although A. breviligulata is generally regarded the stronger competitor, A. arenaria can prevail under low sand supply^{23,30}. The Lévy-type expansion of A. arenaria may explain its efficiency in sand-limited environments, as this strategy may prevent sediment depletion by accreting sand within shoot patches rather than distributing it over a wider area. In contrast, the dispersed A. breviligulata-strategy accretes sand over a wider area, preventing local detrimental effects of excessive sand burial.”

- 12) 3. Relevance of sand trapping efficiency: I don't see why this metric is relevant compared with total sand accretion area. The authors should explain better why total belowground biomass can hinder clonal expansion and thus give a competitive advantage to A. Arenaria.

Reply: We agree that we should have explained better the physiological constraints of growing at sandy beaches and have now included additional analyses of both soil and leaf samples, demonstrating that plants in these environments are nutrient limited (see Supplementary Table 2). We now refer to these nutrient-mediated constraints:

p. 6 L151-156: *“The outcome changes when accounting for the relatively high energy investment of this dispersed strategy, which requires covering long distances relative to more clumping strategies ($\mu > 2$) (Figure 3d). Collected field data suggest that resource efficiency is critical for plants growing in these sandy systems, as the data revealed very low nutrient levels in the soils and leaf tissue of both species (Supplementary Table 2).”*

- 13) 4. Universality of $\mu=2$: Although the introduction of sand trapping efficiency is interesting, the maximum efficiency at $\mu=2$ is completely dependent on the wind and sand accretion models (which contains very strong assumptions and it is quite unrealistic) and there is no reason why that value should be universal. The authors should highlight this fact and modify the conclusions as necessary.

Reply: We understand that the universality of $\mu \sim 2$ seems arbitrary and dependent on the chosen wind and accretion model. However, we found this value to be consistent under a wide range of model simulations. Yet, the number of shoots required to converge to the optimum efficiency of $\mu = 2$ does depend on the conditions. We have now included a sensitivity analysis to illustrate this (see Supplementary Table 3) and clarified our findings in the text:

p.7 L161-163: *“Additional analyses demonstrate that this effect becomes increasingly apparent as the number of shoots in the clonal network increases, although the number of shoots required depends on wind conditions (Supplementary Fig. 7, Supplementary Table 3).”*

- 14) 5. Wind and sand accretion models: The assumptions for the model are quite unrealistic, as the authors use a laminar 2D flow over what is seems like a single rigid cylinder (simulating a single plant shoot), instead of the real 3D turbulent flow over a complex and flexible plant geometry. Also the threshold for sand transport is not really function of wind velocity (which changes with elevation) but wind shear stress within the turbulent boundary layer.

Reply: We of course agree that our biophysical model is a simplified version of a complex natural phenomenon. However, rather than simulating plant growth, sand dynamics, and dune formation in great detail, our model's aim was to highlight that different shoot configurations, generated by different expansion strategies, differ in their

potential to capture sand. Despite its obvious limitations, our model identifies clear organization optima for the overall potential to capture sand, and the efficiency in terms of the distance covered to gain this potential. Moreover, these findings are supported by our field experiment. To clarify the goal (and hence limitation) of our model, we have included the following sentences:

p. 21 L440-445: *“As our aim was to merely examine the effect of shoot organization on wind flow as a proxy for sand capture potential, we constructed a simple model that disregarded many aspects of the complex phenomenon of natural dune formation. In this minimal model, we assume a constant unidirectional flow, no initial beach topography, differences in grain size distribution or sand moisture which are all known to affect transport threshold and shear stress at the sand surface⁵⁴.”*

Furthermore, to further validate our modeling approach we have simulated wind flow over our experimental patterns and the model outcome (both total potential area of sand deposition and sand trapping efficiency) is consistent with the values obtained in the field experiment. We have now included these analyses in our method section:

p.22 L479-484: *“To validate the use of a simplified laminar flow in our biophysical model, we compared the outcome of our field experiment with simulated shoot patterns that reflect the shoot organizations we used in our experiment. We found the results to be consistent, that is, sand deposition was highest in the more dispersed shoot organization whereas sand trapping efficiency was highest in the patchy organization (Supplementary Fig. 8).”*

Indeed, plant geometry is also simplified in our model. However, Zarnetske et al. (*Ecology*, 2002) found the effect of plant morphology to be greatly mitigated by the effect of shoot densities. A simplified plant geometry is a common and successful approach in models dealing with plant-environment feedbacks (Temmerman et al. (*Geology*, 2007), Duran et al. (*Geomorphology*, 2008)). To clarify this assumption in our model we added the following:

p.21 L461-464: *“As the effect of plant morphology on sand capture is greatly mitigated by shoot density²³, we assumed a simple plant geometry in our model and shoot basal area alone was used to characterize the interaction between vegetation and wind flow⁵⁴.”*

- 15) The authors should address the limitations of the model and discuss the dependency of sand deposition area on plant distribution in more general terms. For instance, assuming each shoot has a well defined sheltered area where sand transport is negligible, the results on supplementary figure 6 can be understood as a decrease of total sheltered area (which increases linearly with the number of shoots) with the overlap between shoots. This overlap is minimum when shoots are spread (highest deposition area, low μ) and maximizes for a brownian random walk (lowest deposition area, $\mu \Rightarrow 3$).

Reply: We now clarify the purpose and limitations of the model as described in the previous reply (point 14). Furthermore, we agree that Supplementary Figure 6 (now Supplementary Figure 7) demonstrates the ‘saturating’ effects that a clumping strategy has on potential sand capture. In fact, it highlights why an intermittently clumped ‘Lévy-like’ strategy works well for capturing sand. In a highly dispersed Ballistic ($\mu \sim 1$) strategy, the effect of each individual shoot on wind flow is identical due to the large spacing between shoots. By contrast, in a highly clumped Brownian ($\mu \sim 3$) strategy, the effect of shoots on wind flow indeed strongly overlaps when the plant ages and the number of shoots increases. Our simulations suggest that for sand capture under common wind conditions, it is beneficial to have some overlapping effects in order to

attenuate wind velocity to below the critical velocity threshold for sand deposition. However, once this threshold is overcome by a shoot patch, further expansion increasingly diminishes the additive sand capture potential of new shoots in the patch. This explains why a strategy yielding multiple smaller patches is more effective over time. We now explain this in the text:

p.7 L161-172: *“Additional analyses demonstrate that this effect becomes increasingly apparent as the number of shoots in the clonal network increases, although the number of shoots required depends on wind conditions (Supplementary Fig. 7, Supplementary Table 3). These results demonstrate the saturating effects a clumping strategy ($\mu > 2$) may have on potential sand capture. It therefore highlights why an intermittently clumped ‘Lévy-like’ strategy ($\mu \sim 2$) in early colonization phases (<100 shoots) leads to high potential sand deposition, but on the long run is outcompeted by a more dispersed strategy ($\mu \sim 1.5$) (Supplementary Table 3). Similarly, a highly clumped strategy ($\mu \sim 3$) is more efficient when shoot numbers are low, but as the plant grows, the added attenuating effect of shoots on wind flow decreases due to overlap. Hence, the heavy-tailed ‘Lévy-like’ strategy of $\mu \sim 2$, as observed for *A. arenaria*, becomes more efficient over time by generating multiple shoot patches that maximize engineering effects, while simultaneously colonizing a large area with minimum investment in covering distances.”*

16) 6. Fractal dimension: I don't think it is correct to even define a fractal dimension if the power law scaling is over a range of less than one decade as in supplementary figure 2. In the absence of such a range in the data, the authors should not really invoke any scale invariant or fractal arguments in their results or discussion, unless there are process-based arguments to argue for a scale invariance in plant colonization or clonal expansion.

Reply: We agree that the scale our sampling method offered (up to ~ 1.4 m) is too small to conclude a pattern to be truly self-similar. However, we applied these pattern statistics as a first indication of Lévy-like behavior in the expansion strategy of *A. arenaria* and we further used the fractal dimension as a metric to test whether our random walk model can accurately simulate observed shoot organizations in the field (see Methods section on random walk model). We have modified the text to more explicitly state the scale of similarity observed in our measurements and the purpose of this metric:

p.5 L109-115: *“Spatial cluster analyses revealed that both species strongly deviated from a homogeneous distribution, with *A. arenaria* exhibiting a shoot organization with a fractal dimension of 0.8 over a range of values that our sampling method allowed (4-16 cm) (Supplementary Fig. 2). Since point patterns generated by Lévy movement generally lack a specific scale (Lévy dust)^{27,28}, this provided a first indication that beach grasses seem to diverge from ‘simple’ Brownian movement processes and follow more complex Lévy-like expansion strategies²⁹.”*

17) Minor points:

1. Relation to plant physiology: related to the previous point, there is no discussion of plant physiology even if we can think that ultimately clonal expansion (as related to the spatial and temporal frequency of shoot production) is a product of it. The authors could at least mention some underlying processes behind such strategies and also explore potential reasons why optimal sand accretion could feedback into a particular clonal expansion strategy. After all, the authors seem to suggest the growth of *A. Arenaria* has evolved to maximize the efficiency of sand deposition.

Reply: We agree that the likely ‘optimal’ strategy for either species should be better

explained in the context of their physiology and have modified the text of the introduction to include this.

p. 4 83-88: *“In addition, the plants differ in their physiological tolerance to burial and flooding stress, respectively – with A. arenaria being more resistant to burial stress by developing vertically expanding rhizomes, while A. breviligulata has a higher salinity tolerance. This suggests that both species have adopted different dune-building strategies to cope with the stressful conditions of growing at the land-sea interface²⁶.”*

18) 2. Pareto distribution (Eq. 3): there seems to be a problem with the definition of the distribution as the average value of the step size using Eq.3 diverges for $\mu=2$ instead of $\mu=1$. I think the power in Eq.3 should be $\mu+1$. The authors should confirm this is just typo in the equations.

Reply: It is true that for an unbounded Pareto distribution the average step size approaches infinity as $\mu \rightarrow 2$ (see equation below; Pueyo, *Landscape Ecology*, 2011):

$$\bar{S} = \left(\frac{\mu - 1}{\mu - 2} \right) S_{\min}$$

Therefore, one may argue that people should refrain from fitting unbounded Pareto distributions on empirical data (Pueyo, *Landscape Ecology*, 2011). Nevertheless, especially in the field of movement ecology, many choose to fit probability distributions with unbounded means to describe their empirical data, in addition to the bounded version (e.g. de Jager et al., *Science*, 2011; Edwards et al., *PLoS ONE*, 2012; Huda et al. *Nature Communications*, 2018). Essentially, although the unbounded Pareto distribution does not include an upper bound (maximum step size), it can give an accurate description of the data when the maximum step size is beyond the scale of the actual measurements ($s_{\max} > 1.4\text{m}$). We now clarify our rationale for also including the unbounded Pareto in our analyses:

p.17 L356-363: *“In biology, scale-free properties are confined to a certain spatial range by physical constraints and some people refrain from fitting unbounded Pareto distributions on their data⁴⁹. Nevertheless, the majority of studies on Lévy walk behaviour do include probability distributions with unbounded means to describe their empirical data in addition to bounded distributions^{1,4,46,50}. This is because, although the unbounded Pareto distribution does not include an upper bound (maximum step size), it may provide an accurate enough description of the empirical data when the maximum step size is beyond the scale of measurements (in our case: $s_{\max} > 1.4\text{ m}$).”*

In summary, in the current version, the authors don't provide sufficient evidence to back one of their main conclusion. This should be addressed before the manuscript can be accepted for publication.

Orencio Duran

Reviewer #3 (Remarks to the Author):

19) I. Overview

The authors addressed a very relevant problem – the formation of biogeomorphic landscapes – and connected it to how specific types of plants expand in the space. Their striking result is an observation of a Levy-like movement being performed by those plants, and they connect it to the structure of the sand dune that it forms. I found the reported results very appealing to the Levy flight community, because of the new functional role of this type of strategy. To my knowledge, the results are original. However, I think the work would benefit from reviewing some technical aspects of the analysis and modeling that I judge relevant and will describe below.

Reply: We would like to thank the reviewer for the compliments, and we agree that some of the technical aspects of our work require more clarification. Below we address all specific points in detail.

20) II. General comments

1. Stationary condition. When the plants were selected to be analyzed, how one can infer if the spatial pattern is already stationary? If patterns of different ages were compared, I wonder if one can get heterogeneities that contribute to the Levy pattern statistics due to this effect.

Reply: We analyzed multiple colonizing individual plants and we consistently found Lévy or Composite Brownian distribution to best describe our data (Supplementary Table 1), despite the number of shoots in their rhizomal network, which can be considered a proxy for age. These results demonstrate that the expansion strategy of beach grasses is stationary during early dune development. We now clarify this in the Methods:

p.19 L411-415: *“We consistently found (truncated) Lévy or Composite Brownian to best describe our data, regardless of the number of shoots in the network with a $T\mu = 1.96 \pm 0.06$ for *A. arenaria* and a $T\mu = 1.54 \pm 0.05$ for *A. breviligulata*. We therefore assume that the clonal expansion strategy of beach grasses is stationary during early dune development.”*

21) 2. Box counting. I did not find any reference in the main text to the results of the fractal dimension analysis. Is this analysis relevant to the results, or the Ripley’s K function is sufficient for their claims of heterogeneous spatial distribution?

Reply: We used the fractal dimension in the main text as a first indication of Lévy statistics, and in the methods section to validate the use of a simplified random walk model (which excludes branching and a correlated angle). We have modified the text to clarify this.

p.5 L109-115: *“Spatial cluster analyses revealed that both species strongly deviated from a homogeneous distribution, with *A. arenaria* exhibiting a shoot organization with a fractal dimension of 0.8 over a range of values that our sampling method allowed (4-16 cm) (Supplementary Fig. 2). Since point patterns generated by Lévy movement generally lack a specific scale (Lévy dust)^{27,28}, this provided a first indication that beach grasses seem to diverge from ‘simple’ Brownian movement processes and follow more complex Lévy-like expansion strategies²⁸.”*

22) 3. Typo. There is a typo in Equation (1), the pdf of the exponential distribution. It should

be: $f(s) = \lambda e^{\lambda(s_{\min} - s)}$

Reply: Thank you for notifying us, we have changed the text accordingly.

- 23) 4. Equation (6). From where this expression for estimating the exponent of a TP distribution comes from? It is different from the one that I am aware of (White et al. *Ecology* 89 (2008)).

Reply: The equation used in Table 1 of White et al. *Ecology* (2008) cannot be solved analytically, so in our method section we have formulated the log-likelihood function that can be used to numerically determine the μ that satisfies $dl/d\mu=0$.

- 24) 5. Typo, line 310. The probability density function $g(s)$... The notation used in equation (7) is $f(s)$.

Reply: Thank you for notifying us, we have changed the text accordingly.

- 25) 6. Equation (7). It would be good if the authors could be clearer about how they estimated the parameters for a composite Brownian model. In particular, since you are adding more parameters in this fitting (two exponents plus one weight), how do you penalize your MLE?

Reply: Here we have also numerically derived the parameters by maximizing the log-likelihood function: $L = \sum(g(s))$. We have added this information in the text:

p.18 L386-388: *“The maximum likelihood estimate of the parameters was obtained by maximizing the log-likelihood function $L = \sum_{i=1}^n \log(g(s))$ numerically.”*

MLE is used to estimate the parameters of the probability density functions. The selection of the models is done through comparing the AIC values which penalizes the addition of parameters: $AIC = -2 * L + 2 * k$, where k is the number of parameters (see p. 19 L401-402).

- 26) 7. Equation (8). I did not see the statistics of a lognormal distribution in the table or in the text. I think it should be included in the table, because it is always a *strong* alternative to a Levy model.

Reply: We agree that a lognormal distribution is often a strong alternative. In our case, however, it consistently yielded a poor fit to the data. For clarity and completeness, we have now included the lognormal distribution wAIC value in Supplementary Table 1 and the associated KS statistics.

- 27) 8. Model. About the spatially explicit model: (i) What type of boundary condition (if any) was used in building the spatial location of the steps? (ii) What are the sizes of the maximum step and environment? (The authors only report the minimum size as 0.34 cm..) These parameters are extremely important to interpret the model.

Reply: We used an infinite domain. We have now included this information and the maximum step size in the text:

p. 21 L438-440: *“We explored the effect of differences in clonal expansion strategies (as expressed by their step size distribution) on the potential of an individual clonal plant to capture*

sand with the use of a spatially explicit model in an infinite domain.”

p. 21 L452-454: *“where X is a random uniformly distributed variable ($0 \leq X \leq 1$), s_{min} the minimum step size (set at the minimum step size of our field data: 0.34 cm) and s_{max} the maximum step size (set at the maximum step size: 75.33 cm from our field data).”*

28) 9. Levy dust. There is a key word for the spatial locations visited by a Levy walker (what is plotted in Figure 3); this set of points is called a Levy dust. I suggest the authors to use this key word in their paper because it will draw attention to it.

Reply: We thank the reviewer for this suggestion and have included the term throughout the text.

29) 10. Figure 3. It would be very good if the authors included a scale bar so we can infer the distances in figures a-c.

Reply: Thank you very much for this suggestion, we have now included a scale bar in Figure 3.

30) 11. “However, when μ approaches 1, the movement becomes ballistic as the probabilities of making very large steps or smaller steps become equal”. I find this sentence confusing, and I did a numerical experiment to check it, because I was curious. I generated power-law distributed random numbers with different exponents and checked for the relative orders of magnitude of my samples. I did not get the same proportion of small and large values if $\mu \rightarrow 1$ (I got much more larger values). For the two references cited that I have access (de Jager and Bartumeus), I did not find support for this claim.

Reply: We agree that this sentence was poorly formulated and have clarified our meaning in the text:

p. 16 L350-351: *“However, when μ is very close to 1, the movement becomes ballistic as the probability of making very large steps increases.”*

31) III. Comments regarding the analysis

1. Why the authors did not include in their fitting of statistical models a fit of the minimal step? Since the most important part of identifying Levy flights is the tail of the distribution, I wonder why they decided to apparently fix the minimal step in their analysis. The cited reference (Clauset) has a discussion on why it is necessary to fit the minimal step, and I think the statistical power of the results would benefit from a discussion about the reasons of not including a minimal step fit, or even doing it.

Reply: We have now set a fixed minimum step size at 0.68 cm, as we know from our sampling method that we have a measurement error of 0.34 cm. We therefore took 2x the measurement error as we are unable to distinguish separate shoots this accurate. We prefer this approach over the methods described by Clauset, because it allows us to estimate the best fit to the actual data given the limitations of our sampling method, rather than only looking at the presence of power-law behavior in the tail of the distribution. We have now clarified this in the method section:

p. 15 L324-330: *“Instead of the commonly used approach for estimating the minimum step size for power laws as described in Clauset and co-authors⁴⁸, we adopted a fixed minimum step size, as*

we aimed to identify the distribution function that best fits the majority of our data, rather than identifying power-law behaviour in the tail. To account for the methodological measurement error, calculated from translating pixels to cm (~0.34 cm), we set the minimum step size at twice the error (0.68 cm), as it was not possible to accurately distinguish separate shoots below this minimum value."

32) 2. How many synthetic distributions were generated for the model comparison? I could not find this information in the paper.

Reply: We generated 4 different types of synthetic distributions to validate our random walk model (see methods). We compared the results based on 7 model runs per type and 7 shoot patterns observed in the field. We have now included the number of replicates in the method section.

33) 3. Figure 2. I do not find this cumulative plot very informative. It would be nice to include, for instance, the best fit Levy, composite, and lognormal distributions for both data sets. Maybe split this into two figures, one per type of plant.

Reply: We have included the requested information in Supplementary Fig. 5.

34) 4. One of the main properties of Levy flights statistics is the combination of very long steps and short steps. I suggest the authors to include the information of both the minimum and maximum shoots measured in their statistics table 1, for each plant included in their analysis.

Reply: We have included the requested information in Supplementary Table 1.

35) 5. Supp. Fig. 5: The data shows individual plants in different colors and the combination of all the data in thick markers. In the case of *A. arenaria*, it is evident that the tail statistics (steps larger than 10 cm) is dominated by only a few individuals. Therefore, I think it can be a case that the heterogeneities in the plants are contributing to the power law statistics caveat in the global analysis if the authors simply combine different individuals in the same dataset, because it can be the.

Reply: In fact, all plants had maximum step size larger than 10 cm. As requested, this information is now included in Supplementary Table 1. Please also note that the distributions found for individual plants agree for 10 out of 12 with those on the combined data.

36) Overall conclusion:

As I was reading the paper, I noticed several technical problems on the statistical analysis and missing information about how it was done. Of particular concern is the lack of details in table 1, and also the lack of alternate distribution functions that could be tested (e.g., the lognormal one). I find the works by Humphries et al. particularly good references of how to treat the data in the case of Levy flights (see their Methods in Ecology and Evolution 2013 paper). I understand the technical difficulties of, say, acquiring more data to improve the statistics, but I think that to argue for a Levy-type statistics one has to do an investigation that is deeper than what is currently presented in the paper. I hope my comments will be useful in this direction.

Reply: We have included the requested information in Supplementary Table 1. The aim of our paper was not to find conclusive evidence for Lévy statistics in dune grasses, but

to demonstrate that dune grasses deviate from Brownian movement and that these species-specific heavy-tailed clonal expansion strategies affect potential sand capture during the early colonization phase. Despite the limitations of the data, we found striking differences in expansion strategies between the species, and their resulting sand capture potential.

REVIEWERS' COMMENTS:

Reviewer #2 (Remarks to the Author):

I was kindly asked by the editor to also consider the authors' responses to Reviewer 1's previous concerns. In my opinion, the authors did a good job clarifying the main issues raised by Reviewer 1, in particular by emphasizing that the focus of their work is on the initial stages of dune formation, when plant growth is relatively unaffected by sand erosion or deposition. The authors also responded most of my comments successfully. I now better understand the exploratory character of their work and the fact that both, (1) a causal explanation for the development of a Levy expansion strategy, and (2) the influence of the initial strategy on dune growth beyond the initial phase, are beyond the focus of the present manuscript. I therefore recommend publication of the present manuscript as I consider its main result, even if somehow preliminary, significantly advance the research into the connection between biotic and geomorphic processes.

Reviewer #3 (Remarks to the Author):

The authors answered all the critical points that I found relevant on their statistical analysis of the Levy-like expansion strategy. They added the missing information regarding the fitting of the data and clarified my questions. Therefore, as I can judge for the presented data analysis and interpretation, I am positive that this paper is suitable for publication and will be of great interest to the Levy flight community.

REVIEWERS' COMMENTS:

Reviewer #2 (Remarks to the Author):

I was kindly asked by the editor to also consider the authors' responses to Reviewer 1's previous concerns. In my opinion, the authors did a good job clarifying the main issues raised by Reviewer 1, in particular by emphasizing that the focus of their work is on the initial stages of dune formation, when plant growth is relatively unaffected by sand erosion or deposition. The authors also responded most of my comments successfully. I now better understand the exploratory character of their work and the fact that both, (1) a causal explanation for the development of a Levy expansion strategy, and (2) the influence of the initial strategy on dune growth beyond the initial phase, are beyond the focus of the present manuscript. I therefore recommend publication of the present manuscript as I consider its main result, even if somehow preliminary, significantly advance the research into the connection between biotic and geomorphic processes.

Reviewer #3 (Remarks to the Author):

The authors answered all the critical points that I found relevant on their statistical analysis of the Levy like expansion strategy. They added the missing information regarding the fitting of the data and clarified my questions. Therefore, as I can judge for the presented data analysis and interpretation, I am positive that this paper is suitable for publication and will be of great interest to the Levy flight community.

Reply: We thank the reviewers for their useful comments and were happy to read that they were satisfied with our revisions.